METHODS AND RESOURCES

# A sensitive soma-localized red fluorescent calcium indicator for in vivo imaging of neuronal populations at single-cell resolution

Shihao Zhou[1,2,3☯], Qiyu Zhu[4,5,6☯], Minho Eom[7☯], Shilin Fang[8☯], Oksana M. Subach[9], Chen Ran[10], Jonnathan Singh Alvarado[11], Praneel S. Sunkavalli[11], Yuanping Dong[1,2], Yangdong Wang[1,2,3], Jiewen Hu[8], Hanbin Zhang[1,2,3], Zhiyuan Wang[8,12], Xiaoting Sun[1,2,3], Tao Yang[1,2], Yu Mu [8,12]*, Young-Gyu Yoon[7,13,14]*, Zengcai V. Guo[4,5,6]*, Fedor V. Subach[9]*, Kiryl D. Piatkevich [1,2,3]*

1 School of Life Sciences, Westlake University, Hangzhou, Zhejiang, China, 2 Westlake Laboratory of Life Sciences and Biomedicine, Hangzhou, Zhejiang, China, 3 Institute of Basic Medical Sciences, Westlake Institute for Advanced Study, Hangzhou, Zhejiang, China, 4 IDG/McGovern Institute for Brain Research, Tsinghua University, Beijing, China, 5 School of Basic Medical Sciences, Tsinghua University, Beijing, China, 6 Tsinghua-Peking Center for Life Sciences, Beijing, China, 7 School of Electrical Engineering, KAIST, Daejeon, Republic of Korea, 8 Institute of Neuroscience, State Key Laboratory of Neuroscience, Center for Excellence in Brain Science and Intelligence Technology, Chinese Academy of Sciences, Shanghai, China, 9 Complex of NBICS Technologies, National Research Center "Kurchatov Institute", Moscow, Russia, 10 Department of Neuroscience, Dorris Neuroscience Center, The Scripps Research Institute, La Jolla, California, United States of America, 11 Division of Endocrinology, Diabetes and Metabolism, Beth Israel Deaconess Medical Center, Harvard Medical School, Boston, Massachusetts, United States of America, 12 University of Chinese Academy of Sciences, Beijing, China, 13 KAIST Institute for Health Science and Technology, Daejeon, Republic of Korea, 14 Department of Semiconductor System Engineering, KAIST, Daejeon, Republic of Korea

☯ These authors contributed equally to this work.
* kiryl.piatkevich@westlake.edu.cn (KDP); subach_fv@nrcki.ru (FVS); guozengcai@mail.tsinghua.edu.cn (ZVG); younggyu.yoon@gmail.com (YGY); my@ion.ac.cn (YM)

## Abstract

Recent advancements in genetically encoded calcium indicators, particularly those based on green fluorescent proteins, have optimized their performance for monitoring neuronal activities in a variety of model organisms. However, progress in developing red-shifted GECIs, despite their advantages over green indicators, has been slower, resulting in fewer options for end users. In this study, we explored topological inversion and soma-targeting strategies, which are complementary to conventional mutagenesis, to re-engineer a red genetically encoded calcium indicator, FRCaMP, for enhanced in vivo performance. The resulting sensors, FRCaMPi and soma-targeted FRCaMPi (SomaFRCaMPi), exhibit up to 2-fold higher dynamic range and peak $\Delta F/F_0$ per single AP compared to widely used jRGECO1a in neurons both in culture and in vivo. Compared to jRGECO1a and FRCaMPi, SomaFRCaMPi reduces erroneous correlation of neuronal activity in the brains of mice and zebrafish by two- to 4-fold due to diminished neuropil contamination without compromising the signal-to-noise ratio. Under wide-field imaging in primary somatosensory and visual cortices in mice with high labeling density (80–90%), SomaFRCaMPi exhibits up to

**Data availability statement:** All relevant data are within its Supporting Information files. Custom code used in this study is available at https://github.com/Shihao-Neuro/SomaFRCaMPi and also via Zenodo at https://doi.org/10.5281/zenodo.14942673

**Funding:** The work was supported by the National Natural Science Foundation of China grant 32171093 (K.D.P) (https://www.nsfc.gov.cn/english/site_1/index.html); 'Pioneer' and 'Leading Goose' R&D Program of Zhejiang 2024SSYS0031 (K.D.P) (https://kjt.zj.gov.cn/); Foundation of Westlake University, Westlake Laboratory of Life Sciences and Biomedicine (K.D.P) (https://en.wllsb.edu.cn/). The work was also carried out within the state assignment of NRC "Kurchatov institute" (design and FRCaMPi characterization in vitro) (F.V.S). The work was also supported by the Ministry of Science and Higher Education of the Russian Federation for the development of the Kurchatov Center for Genome Research 075-15-2019-1659 (FRCaMPi characterization in neurons) (http://nrcki.ru/); National Natural Science Foundation of China grant 32170998 (Z.V.G) (https://www.nsfc.gov.cn/english/site_1/index.html); The Creative Research Groups of the National Natural Science Foundation of China, 32321003 (Y.M) (https://www.nsfc.gov.cn/english/site_1/index.html); National Science and Technology Innovation 2030 Major Program 2021ZD0204502, 2021ZD0203704 (Y.M) (https://www.ncsti.gov.cn/); Simons Collaboration on the Global Brain and NIH K01 DK132957 (C.R) (https://www.simonsfoundation.org/); NIH NEI Grant 1 S10 OD026817-01 (C.R) (https://www.nei.nih.gov/grants-and-training/funding-opportunities/current-funding-opportunities); Harvard Brain Initiative Pioneer Award (J.S.A) (https://brain.harvard.edu/grants/postdoc-pioneers-grant-program/). The funders did not play any role in the study design, data collection and analysis, decision to publish, or preparation of the manuscript.

**Competing interests:** The authors have declared that no competing interests exist.

**Abbreviations:** CeA, central amygdala; cpRFP, circularly permutated red fluorescent proteins; dpf, days post-fertilization; FOV, field of view-GECIs, genetically encoded calcium indicators;

40% higher SNR and decreased artifactual correlation across neurons. Altogether, SomaFRCaMPi improves the accuracy and scale of neuronal activity imaging at single-neuron resolution in densely labeled brain tissues due to a 2–3-fold enhanced automated neuronal segmentation, 50% higher fraction of responsive cells, up to 2-fold higher SNR compared to jRGECO1a. Our findings highlight the potential of SomaFRCaMPi, comparable to the most sensitive soma-targeted GCaMP, for precise spatial recording of neuronal populations using popular imaging modalities in model organisms such as zebrafish and mice.

## Introduction

Genetically encoded calcium indicators (GECIs) have been long established as a standard tool for real-time recording of activities from neuronal compartments and neuron populations in vivo in model organisms [1–5]. In recent years, GECIs based on green fluorescent proteins have received substantial improvements in their sensitivity (i.e., $\Delta F/F_0 = (F_{max} - F_0)/F_0$, defined as maximal fluorescence change over the baseline of the calcium transient), which is crucial in defining performance of the sensors for in vivo imaging [6]. These advancements in sensitivity have been achieved through iterative mutagenesis, as seen in the GCaMP series [2–4,7,8], or by increasing the dynamic range of the indicators with a brighter fluorescent protein (for example, mNeonGreen instead of GFP in NCaMP7 and NEMOs) [9,10].

Despite the advancement in engineering green GECIs, creating calcium sensors with red-shifted fluorescence spectra offers certain advantages over green ones, and enables spectral multiplexing with a variety of already available green indicators [5,11–13]. After engineering the first red-shifted Ca$^{2+}$ indicator, R-GECO1 [14], several red Ca$^{2+}$ indicators based on circularly permutated red fluorescent proteins (cpRFP) have been developed with improved sensitivity [13,15–17]. However, progress in developing and optimizing red-shifted GECIs has been much slower and more challenging than green ones, with limited variants [18] and inferior in vivo performance. Similar to GCaMPs, advancements in red GECIs also often come at the cost of extensive directed molecular evolution followed by time-consuming and laborious characterization of the top variants to select one that is best performing for a particular application.

Alternative strategies to extend the applicability of genetically encoded indicators are engineering topological variants and cellular focusing by targeting specific subcellular compartments, e.g., nucleus and soma. The topological engineering reverses the conventional GCaMP-like design to a so-called non-circularly permutated (ncp) topology with C- and N-terminus reassigned to the FP, similar to the topology of GECIs such as NCaMP7 (ref. [10]) and camgaroo [1]. One of the major outcomes of the topological inversion of the existing indicators is an elevated ligand binding affinity [19,20], which was recently shown to boost the amplitude of signal response [21]. However, this engineering strategy has been rarely explored in the existing red GECIs [21]. Complementary to topological engineering, soma-targeting

HBSS, Hanks balanced salt solution; IACUC, institutional animal care and use committee; IOU, intersection-over-union; ncp, non-circularly permutated; NES, nuclear exclusion sequence; NTS, nucleus of the solitary tract; PBS, phosphate-buffered saline; PCC, Pearson correlation coefficient; PFA, paraformaldehyde; RGC, retinal ganglion cells; ROI, region of interest; RPL10, ribosome protein L10; SNR, signal-to-noise ratio; 1-FP, single fluorescent protein.

of GECIs is known to improve the effectiveness of population calcium imaging at single-neuron resolution in vivo due to reduced neuropil contamination, as was demonstrated specifically for wide-field imaging, one of the most popular imaging modalities, in zebrafish and mice (see text in S1 Text more in-depth discussion). Through fusion with soma-localization motif appended to the C-terminus of sensors, such strategy facilitates image segmentation for extraction of single cell traces, reduces signal contamination from neighboring neuropils [22,23], and increases signal-to-noise ratio (SNR) [23]. However, this strategy has not been tested to improve the previous and newly generated red GECIs.

Here, we employed these strategies to generate FRCaMPi and SomaFRCaMPi, a pair of red GECIs that demonstrate high dynamic range with enhanced sensitivity. FRCaMPi is the inverted topological variant of FRCaMP, while SomaFRCaMPi is the first reported soma-localized calcium indicator in red, which was engineered by tethering FRCaMPi to the 60S subunit of ribosome via ribosome protein L10 (RPL10). Both FRCaMPi and SomaFRCaMPi were evaluated in vivo in zebrafish larvae using confocal imaging and in awake, behaving mice using wide-field, two-photon imaging, and fiber photometry. Compared to conventional red GECIs under identical imaging conditions, SomaFRCaMPi minimized neuropil contamination and further demonstrated greater performance in metrics relevant to imaging neural populations in both wide-field and two-photon setups in vivo. Importantly, SomaFRCaMPi outperformed soma-localized versions of all other state-of-the-art red GECIs generated in this study along all major axes of performance.

## Results

### Engineering and biochemical characterization of FRCaMPi

The NCaMP7-like topological design [10], which involves inserting the sensing domains within the FP, has recently resurged in the field of fluorescent biosensors and resulted in the successful construction of several green ncp indicators for glutamate and calcium. Those examples include ncpiGluSnFRs [19] for glutamate, ultrahigh-affinity GECIs [21] for detecting low nanomolar $Ca^{2+}$ levels (<100 nM), bioluminescent $Ca^{2+}$ indicator based on ncpGCaMP6s [20], and mNeonGreen-based $Ca^{2+}$ indicators of superior sensitivity [9,10]. Recently, FRCaMP was introduced as the first red GECI engineered based on the calcium-binding domain CaM and the M13 peptide from a non-metazoan origin, possessing a similar dynamic range but a greater sensitivity than R-GECO1 [24]. In addition, FRCaMP did not interact with the cytosolic environment of the mammalian cells. Motivated by these advancements, we initially explored the benefits of an altered topology in FRCaMP. To alter the topology of FRCaMP, we genetically connected the terminal of M13 and CaM with a flexible linker (GGS-GGSGSGS) and reintroduced the original termini of mApple protein (Fig 1A). We then termed this ncpFRCaMP as FRCaMP with the inverted topology (FRCaMPi; i stands for "inverted"). Consistent with previous observations of the biochemical properties of ncp indicators, in vitro characterization revealed that FRCaMPi binds $Ca^{2+}$ with a dissociation constant ($K_d$) that is approximately 3-fold lower (81 ± 2 nM) compared to FRCaMP (214 ± 6 nM; S3 Table and S1D Fig). Similarly, FRCaMPi inherited the

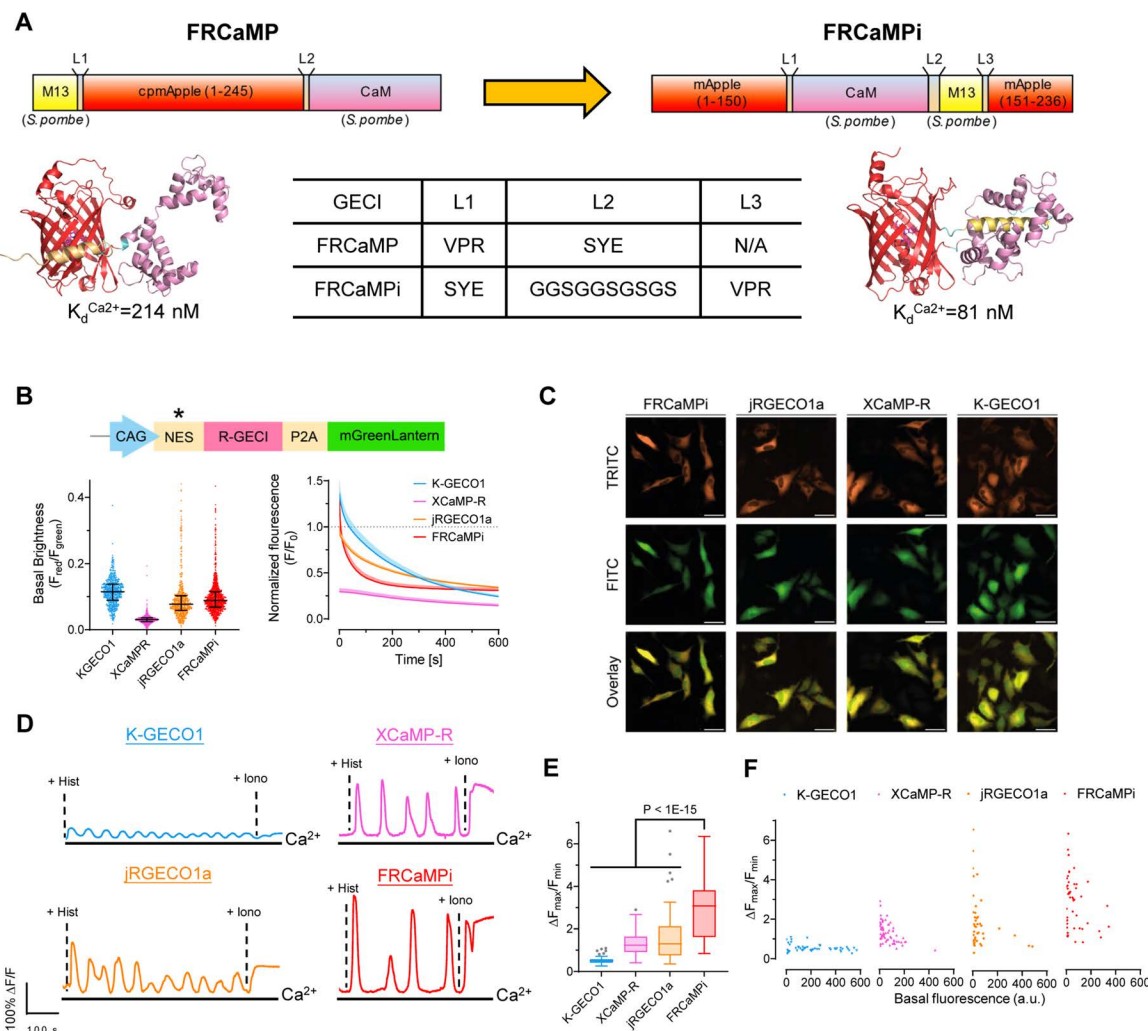

**Fig 1. Design of FRCaMPi and characterization in HeLa cells. (A)** FRCaMPi sensor is created by topological inversion of FRCaMP. Upper panel, a diagram showing the design of FRCaMP and FRCaMPi; the lower panel predicted 3D protein structures of indicators75 and a table of amino acid sequences of the linkers. **(B)** Basal brightness and photobleaching curves of red GECIs characterized in transfect HeLa cells by side-by-side comparison. For the plasmids used to transfect HeLa cells, a nuclear exclusion sequence (NES) was added to each red GECI and mGreenLantern reference was attached via a P2A peptide to express red and green FP in near equimolar ratio. *NES was not added to XCaMP-R due to the presence of F2A sequence C-terminal reported to have nuclear exclusion. The basal fluorescence of red GECIs ($F_{red}$) was normalized against the fluorescence of mGreenLantern ($F_{green}$). Fluorescence trace of FRCaMPi, jRGECO1a, XCaMP-R, and K-GECO1 over 10-min photobleaching, normalized to the mean brightness of jRGECO1a in the first frame. **(C)** Representative images showing the expression of mGreenLantern and FRCaMPi in HeLa cells. **(D)** Representative fluorescence trace induced by histamine (50 μM) and ionomycin (5 μM) in HeLa cells expressing K-GECO1, XCaMP-R, jRGECO1a, and FRCaMPi. **(E)** $\Delta F_{max}/F_{min}$ of FRCaMPi compared to other red GECIs during the period of pharmaceutically induced Ca$^{2+}$ response (K-GECO1: $n$ = 53 cells from 4 wells; XCaMP-R: $n$ = 71 cells from 5 wells; jRGECO1a: $n$ = 50 cells from 4 wells; FRCaMPi: $n$ = 50 cells from 4 wells, two independent cultures). Kruskal–Wallis multiple-comparison test: $p < 1 \times 10{-15}$. Pairwise Dunn's comparison test with FRCaMPi: $p$ = 5.92e–5 (jRGECO1a), $p$ = 9.73e–7 (XCaMP-R) and $p < 1$e–15 (K-GECO1). **(F)** Scatter plot of $\Delta F_{max}/F_{min}$ against basal brightness of individual cells. Statistics were the same as **(E)**. The quantitative data presented in this figure can be found in S1 Data.

outstanding dynamic range (16.3 ± 0.3-fold) from FRCaMP (16.4 ± 0.7). The Hill coefficient for FRCaMPi was slightly higher (3.1 ± 0.2) compared to that of FRCaMP (2.5 ± 0.2) (S3 Table). In addition, the biophysical properties of FRCaMPi, such as fluorescence spectra, quantum yield, extinction coefficient, dynamic range, and p$K_a$, remained very similar to the parental FRCaMP (S1B and S1C Fig and S3 Table).

The increased calcium binding affinity and preserved key features from FRCaMP, such as large dynamic range, led us to compare the performance of FRCaMPi directly with FRCaMP in cultured neurons using external electrical bursts at 87 Hz of 300 APs to evoke Ca²⁺ responses. Notably, FRCaMPi demonstrated a peak fluorescence response that was 2.36 times as high as that observed with FRCaMP (S2B Fig). We found that the decay half kinetics is slightly slower for FRCaMPi in comparison to FRCaMP (S2C Fig). Sensors featured by a higher calcium binding affinity are known to buffer a greater quantity of Ca²⁺ and could result in heightened sensitivity but prolonged decay kinetics [2,3,6,8,21]. This was consistent with our observation in FRCaMPi. In conclusion, topological inversion of FRCaMP resulted in a new sensor with over twice the sensitivity of its precursor, thus representing a promising candidate for further applications in neurons both in vitro and in vivo.

## Characterization of FRCaMPi in mammalian cell cultures

Since FRCaMPi demonstrated even higher sensitivity in the initial in vitro characterization, we decided to benchmark FRCaMPi performance against other known state-of-the-art red calcium indicators, including X-CaMPR [16], K-GECO1 [15], jRGECO1a [13] (Table 1), characterized by high sensitivity in the cytosol of mammalian cells. To demonstrate the utility of FRCaMPi in mammalian cells, we first characterized the indicator in cultured HeLa cells. By expressing a bicistronic vector, the basal brightness of red GECI was reported as the fluorescence ratio of the indicator fluorescence to the fluorescence of a reference protein, mGreenLantern (Fig 1B and 1C). A similar brightness level was found between FRCaMPi and jRGECO1a and the brightness level of K-GECO1 was slightly higher (1.2-fold) compared to FRCaMPi (Fig 1B). Under continuous wide-field illumination at 7.4 mW/mm² (about 4–5 times higher than typically used for FRCaMPi imaging in cultured cells), the photobleaching rate of FRCaMPi was about 2- and 3.6-folds faster than those of jRGECO1a and K-GECO1, respectively. Nevertheless, we observed a quicker signal plateau of FRCaMPi after 3 min with eventually a brightness level similar to jRGECO1a at 10 min of bleaching time (Fig 1B).

To assess the capability of FRCaMPi to detect Ca²⁺ signals in HeLa cells, we recorded the fluorescent responses to cytoplasmic Ca²⁺ transients elicited by consecutive application of histamine followed by ionomycin. During pharmacologically induced Ca²⁺ changes, the maximal fluorescence response ($\Delta F_{max}/F_{min}$) of HeLa cells expressing FRCaMPi was at least 2-fold to the response of XCaMP-R and jRGECO1a and was more than 4 folds higher than that for K-GECO1 (Fig 1D and 1E and S1 Movie). To this point, we noted that FRCaMPi exhibited a larger dynamic range without sacrificing a baseline

**Table 1. In vitro biophysical properties of red GECIs.**

| Name of GECI | Ex/Em (nm) apo | Ex/Em (nm) sat | Quantum yield | ε (mM⁻¹ cm⁻¹) | pKa | Dynamic range | Kd (nM) | Hill coefficient | References |
|---|---|---|---|---|---|---|---|---|---|
| K-GECO1 | 568 (−) | 565 (−) | 0.12 (−) | 19 (−) | / | 12 | 165 | 1.12 | Shen et al. (2018) [15] |
| | 594 (+) | 590 (+) | 0.45 (+) | 61 (+) | / | | | | |
| XCaMP-R | 574 (−) | 561 (−) | 0.14 (−) | 37.7 (−) | 6.06 ± 0.03 (−) | 5.57 | 97 ± 10 | 1.1 ± 0.1 | Inoue et al. (2019) [16] |
| | 598 (+) | 593 (+) | 0.28 (+) | 76.3 (+) | 8.72 ± 0.02 (+) | | | | |
| jRGECO1a | 577 (−) | 562 (−) | 0.12 (−) | 6.18 (−) | 8.6 (−) | 11.6 ± 0.4 | 148 ± 2 | 1.90 ± 0.02 | Dana et al. (2016) [13] |
| | 595 (+) | 588 (+) | 0.22 (+) | 53.3 (+) | 6.3 (+) | | | | |
| FRCaMPᵃ | 576 (−) | 564 (−) | 0.13 (−) | 26.16 ± 0.08 (−) | 8.88 ± 0.05 (−) | 16.4 ± 0.7 | 214 ± 6 | 2.5 ± 0.2 | Subach et al. (2021) [24] |
| | 602 (+) | 592 (+) | 0.23 (+) | 53 ± 2 (+) | 6.60 ± 0.04 (+) | | | | |
| FRCaMPiᵃ | 576 (−) | 566 (−) | 0.12 (−) | 27.89 ± 0.08 (−) | 8.98 ± 0.07 (−) | 16.3 ± 0.3 | 81 ± 2 | 3.1 ± 0.2 | This paper |
| FRCaMPiᵃ | 576 (−) | 566 (−) | 0.12 (−) | 27.89 ± 0.08 (−) | 8.98 ± 0.07 (−) | 16.3 ± 0.3 | 81 ± 2 | 3.1 ± 0.2 | This paper |
| | 600 (+) | 594 (+) | 0.23 (+) | 50.2 ± 3.3 (+) | 6.48 ± 0.02 (+) | | | | |

ᵃ$K_d$ and Hill coefficient value of FRCaMP and FRCaMPi were reported using purified protein solution in the absence of Mg²⁺.

PLOS Biology

fluorescence. In support of this notion, we also found that in pharmaceutically induced $Ca^{2+}$ responses, cells expressing FRCaMPi showed matched basal fluorescence levels of those expressing jRGECO1a still exhibited a distribution over higher dynamic range, indicating a higher maximal brightness of FRCaMPi that can possibly achieve (Fig 1F).

Next, we systematically and quantitatively assessed $\Delta F/F_0$, SNR, and kinetics of response of FRCaMPi in cultured mouse hippocampal neurons, adapting an electrical stimulation protocol that was well established for other GECIs, including GCaMPs, jRGECO1a, and NIR-GECO1 [2,5,13] (Figs 2A and S3A-S3C). Field stimuli (45–50 V, 83 Hz, 1 ms) were delivered in trains of 1, 2, 3, 5, 10, 20, 40, 80, and 160 pulses to transduced mouse hippocampal neurons and 5 min of

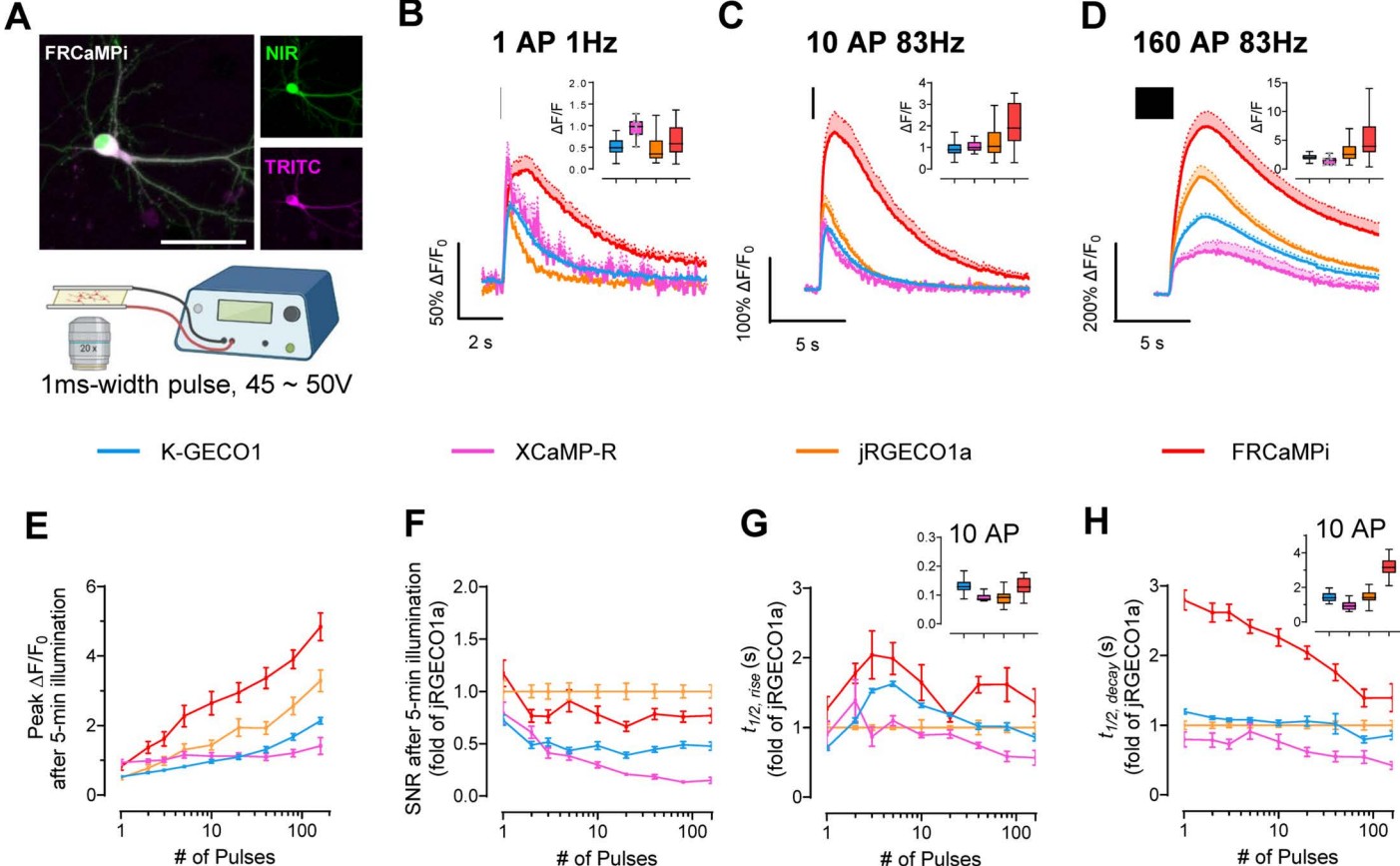

**Fig 2. Characterization of FRCaMPi in primary mouse hippocampal neurons. (A)** FRCaMPi sensor is created by topological inversion of FRCaMP. Upper panel, a diagram showing the design of FRCaMP and FRCaMPi; the lower panel predicted 3D protein structures of indicators 75 and a table of amino acid sequences of the linkers. Representative images of a neuron co-expressing FRCaMPi (TRITC) and emiRFP670 (NIR) (top) and schematics of field stimulation protocol (*n* = 5 neurons from two independent cultures). Scale bar, 50 μm. **(B–D)** Averaged fluorescence traces generated in response to **(B)** one pulse, **(C)** 10 pulses, and **(D)** 160 pulses for tested GECIs. Traces are shown as mean ± s.e.m, indicated by solid lines and shaded area, respectively (K-GECO1, *n* = 54 cells from 3 coverslips, 2 independent cultures; XCaMP-R, *n* = 24 cells from 2 coverslips, 2 independent cultures; jRGECO1a, *n* = 65 cells from 4 coverslips, 3 independent cultures; FRCaMPi, *n* = 56 cells from 6 coverslips, 3 independent cultures). Graph inset displays boxplot of peak $\Delta F/F_0$ of each GECI to applied action potentials. **(E)** Peak $\Delta F/F_0$ of FRCaMPi compared to other red GECIs as a function of pulses. Descriptive statistics here and for panels g,h,i same as in **C–E**. **(F)** Normalized SNR of FRCaMPi compared to other red GECIs as a function of pulses. Normalized to the mean value of jRGECO1a at each used pulse number. **(G)** The half the rise time of red GECIs as a function of pulses. (**I**) Half decay time of red GECIs as a function of pulses. Graph inset displays boxplot of half rise/decay time of each GECI after 10 action potentials. Box plots are used here (insets in panels **C**, **D**, **E**, **H**, **I**). Box indicates the median and 25–75th percentile range, and the whiskers represent 1.5 times the interquartile range. Line plots are used here (**E–H**) and throughout the paper (dot, mean; error bars, s.e.m.) See S4 Table for detailed statistics and exact *p* values. The quantitative data presented in this figure can be found in S1 Data. Created in BioRender, https://BioRender.com/e01g235.

illumination were applied prior to trace recording to remove initial bleaching to equalize the effect of bleaching to all indicators (S4E Fig). Fluorescence changes detected from cell bodies show that FRCaMPi possesses a 50–60% larger peak $\Delta F/F_0$ compared to jRGECO1a and K-GECO1 at 1 pulse (Fig 2B and 2E and Table 2). As the number of electrical pulses increased, the peak difference of FRCaMPi to K-GECO1 and XCaMP-R became more distinct (2.77 and 3.34 folds, respectively) and stayed approximately 1.5 times as high as jRGECO1a (Fig 2B, 2C and 1E and Table 2). GECIs with a high calcium affinity and Hill coefficient tend to saturate more easily at smaller spikes [6]. However, despite possessing a low $K_d$ and a Hill coefficient larger than 3, fluorescence responses of FRCaMPi showed fair linearity on the log scale with no notable saturation up to 160 pulses (Fig 2D and 2E). This behavior could be attributed to the extensive dynamic range of the indicator, as supported by experiments in HeLa cell calcium responses. The baseline brightness of FRCaMPi in primary mouse hippocampal neurons observed was not significantly different from that of jRGECO1a but higher than that of X-CaMPR and K-GECO1 (S4C Fig). Given a similar level of baseline brightness but more rapid bleaching, the peak SNR of FRCaMPi remains comparable to jRGECO1a after 5 min of illumination (Fig 2F and Table 2), which could be attributed by the larger sensitivity (i.e., peak $\Delta F/F_0$ in response to pulse trains) of the indicator. The rise kinetics of FRCaMPi appeared similar to those of other indicators (Fig 2G and Table 2), while the decay kinetics of FRCaMPi was slower, with the 2.8-fold difference to jRGECO1a decreased over the increased number of pulses (Fig 2H and Table 2). Altogether, FRCaMPi faithfully reports $Ca^{2+}$ dynamics in mammalian cell cultures with greater dynamics and sensitivity observed than existing red indicators of high sensitivity.

Red genetically encoded calcium indicators (GECIs) are known to have significantly compromised functional performance in vivo, particularly during long-term expression. Therefore, before proceeding with further development, we assessed the fluorescence response of FRCaMPi expressed in the central amygdala (CeA) neurons using fiber photometry (S5A-C Fig). Notably, even after three months of expression, FRCaMPi exhibited reproducible and reliable responses during licking behavior and air-puff stimulus (S5D and S5E Fig). The combined in vitro and in vivo results thus encouraged us to optimize the probe further for its soma-targeted version.

## Development and characterization of soma-targeted FRCaMPi

A recent effort in developing soma-localized green GECIs [22,23] has inspired us to explore the possibility of engineering a competent red soma-localized counterpart. To achieve this, we appended C-terminus of FRCaMPi with the four peptides known to facilitate somatic localization and assessed them in cultured neurons for fluorescence changes (peak $\Delta F/F_0$), SNR in response to evoked extracellular stimulation, and cytotoxicity (assessed as total number of functional cells obtained from transfected neuronal culture; S6 Fig). Based on the screening, we chose to use the RPL10-tagged

**Table 2. Properties of red GECIs measured in dissociated neurons.**

| Name of GECI | 1 AP $\Delta F/F_0$ | 10 AP $\Delta F/F_0$ | 160 AP $\Delta F/F_0$ | 1 AP SNR | 10 AP SNR | 1 AP half-rise time (ms) | 1 AP half-decay time (s) |
|---|---|---|---|---|---|---|---|
| K-GECO1 | 0.53 ± 0.03 | 0.97 ± 0.06 | 2.13 ± 0.09 | 31.12 ± 2.09 | 66.98 ± 5.08 | 91.4 ± 4.58 | 1.13 ± 0.03 |
| XCaMP-R | 0.94 ± 0.08 | 1.12 ± 0.1 | 1.42 ± 0.24 | 61.48 ± 6.23 | 33.97 ± 4.47 | 116.05 ± 25.2 | 0.75 ± 0.1 |
| jRGECO1a | 0.5 ± 0.06 | 1.44 ± 0.14 | 3.29 ± 0.31 | 42.83 ± 4.28 | 139.28 ± 9.34 | 128.84 ± 6.29 | 0.95 ± 0.06 |
| FRCaMPi | 0.82 ± 0.1 | 2.65 ± 0.34 | 4.84 ± 0.4 | 50.19 ± 5.44 | 108.65 ± 11.2 | 163.02 ± 23.13 | 2.65 ± 0.13 |
| SomaK-GECO1 | 0.23 ± 0.02 | 0.32 ± 0.03 | 0.38 ± 0.02 | 14.8 ± 1.18 | 15.14 ± 1.51 | 92.68 ± 8.01 | 1.29 ± 0.15 |
| SomaXCaMP-R | 0.58 ± 0.06 | 1.2 ± 0.07 | 1.84 ± 0.16 | 18.08 ± 1.06 | 26.45 ± 1.11 | 113.19 ± 11.91 | 0.76 ± 0.04 |
| SomajRGECO1a | 1.08 ± 0.1 | 1.97 ± 0.16 | 2.85 ± 0.42 | 21.79 ± 1.27 | 29.63 ± 1.51 | 192.95 ± 18.74 | 0.86 ± 0.05 |
| SomaFRCaMPi | 1.74 ± 0.07 | 3.62 ± 0.17 | 5.36 ± 0.25 | 41.85 ± 1.82 | 74.79 ± 3.32 | 176.61 ± 20.9 | 1.95 ± 0.06 |

Data were obtained from images of stimulated neurons after 5 min of baseline recording.

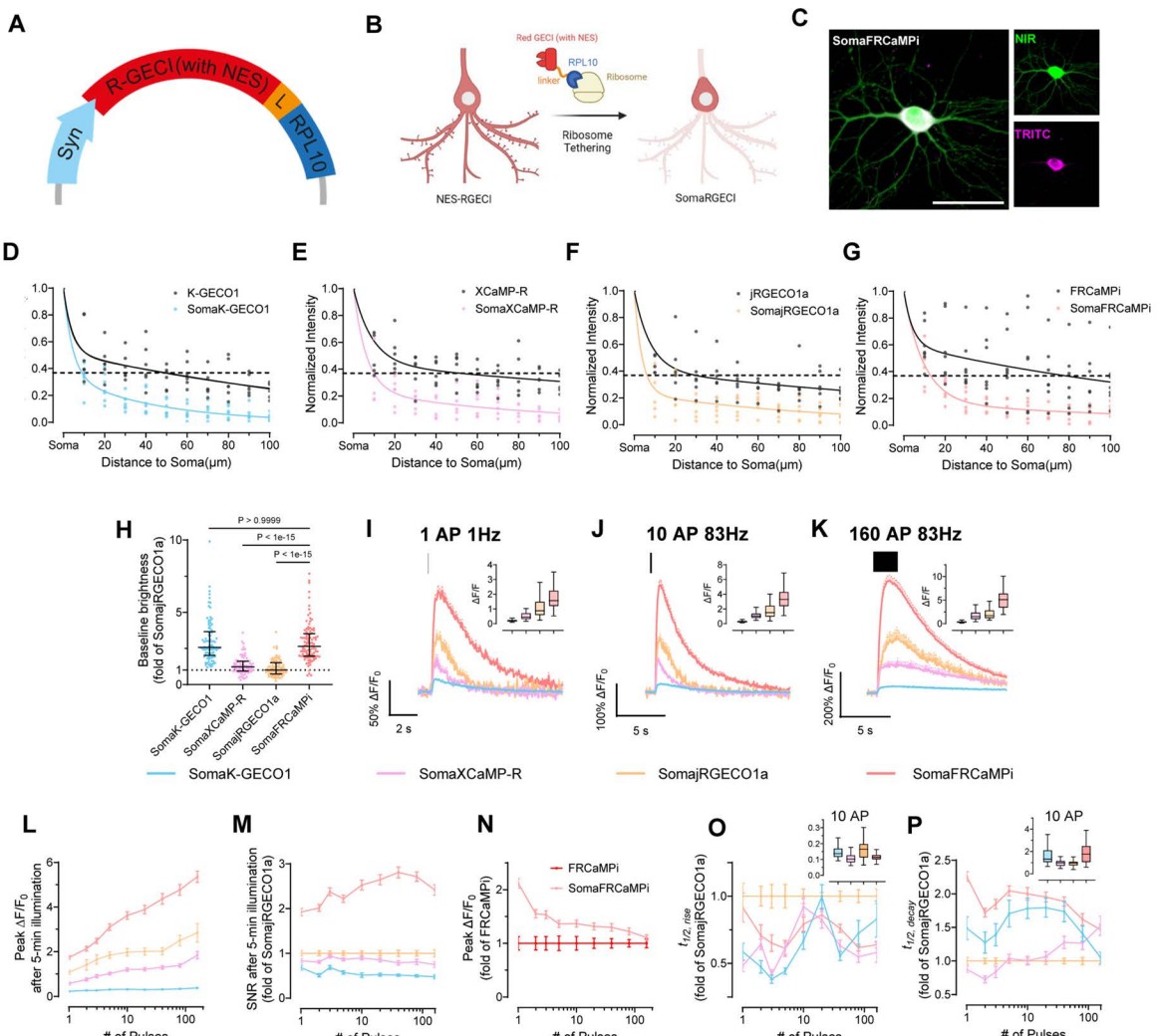

**Fig 3. SomaFRCaMPi engineering and characterization via electric field stimulation in primary mouse hippocampal neurons. (A, B)** Schematics of expression cassette and neurons showing the engineering of soma-localized red GECIs by tethering red GECI to ribosome via rpl10a peptide. **(C)** Representative images of a neuron co-expressing SomaFRCaMPi (TRITC) and emiRFP670 (NIR) ($n = 5$ from two independent neuronal cultures). Scale bar, 50 μm. **(D–G)** Plot of fluorescence intensity along neurites, normalized to the intensity at the soma, from neurons expressing either **(D)** K-GECO1 or SomaK-GECO1, **(E)** XCaMP-R or SomaXCaMP-R, **(F)** jRGECO1a or SomajRGECO1a, **(G)** FRCaMPi or SomaFRCaMPi (K-GECO1: $n = 6$ neurites from 5 cells; SomaK-GECO1: $n = 7$ neurites from 5 cells; XCaMP-R: $n = 6$ neurites from 5 cells; SomaXCaMP-R: $n = 6$ neurites from 4 cells; jRGECO1a: $n = 6$ neurites from 5 cells; SomajRGECO1a: $n = 6$ neurites from 5 cells; FRCaMPi: $n = 7$ neurites from 6 cells; SomaFRCaMPi: $n = 6$ neurites from 6 cells). An unpaired student *t*-test was used. Dots, individual neurites; solid line, exponential fitting curve. The data of the non-targeted indicator was colored in gray to aid visualization. **(H)** Baseline brightness of soma-localized red GECIs characterized in primary hippocampal neurons by side-by-side comparison (SomaK-GECO1: $n = 90$ cells from 5 wells; SomaXCaMP-R: $n = 84$ cells from 4 wells; SomajRGECO1a: $n = 85$ cells from 5 wells; SomaFRCaMPi: $n = 101$ cells from 5 wells, two independent cultures). Kruskal–Wallis ANOVA was used followed by a post hoc Dunn's multiple comparison test. Dot plot is used here and throughout the paper (dots, individual neurons; horizontal line, median; whiskers, interquartile range). Data is normalized to the median of the SomajRGECO1a values. **(I–K)** Averaged $\Delta F/F$ Ca²⁺ transients to one pulse **(I)**, 10 pulses **(J)**, and 160 pulses **(K)** by GECIs. All averaged traces are shown as mean ± s.e.m, indicated by solid lines and shaded area, respectively. Graph inset displays boxplot of peak $\Delta F/F_0$ of each GECI to applied action potentials. Statistics in **C–E** are the same as in **F**. **(L)** Peak $\Delta F/F_0$ of SomaFRCaMPi compared to other red GECIs as a function of pulses. **(M)** SNR of FRCaMPi compared to other red GECIs as a function of pulses. **(N)** Peak $\Delta F/F_0$ of SomaFRCaMPi compared to that of FRCaMPi as a function of pulses. Statistics in o for SomaFRCaMPi and FRCaMPi are the same as in **J** and Fig 1F, respectively. **(O)** Half the rise time of red GECIs as a function of pulses. **(P)** Half decay time of red GECIs as a function of pulses (SomaK-GECO1: $n = 61$ cells from 4 coverslips, 3 independent cultures; SomaXCaMP-R: $n = 64$ cells from 4 coverslips, 3 independent cultures; SomajRGECO1a: $n = 100$ cells from 6 coverslips, 4 independent cultures; SomaFRCaMPi: $n = 148$ cells from 7 coverslips, 4 independent cultures). Graph inset displays boxplot of half rise/decay time of each GECI after 10 action potentials. Box indicates the median and 25–75th percentile range, and the whiskers represent 1.5 times the interquartile range. See S5 Table for detailed statistics and exact *p* values. The quantitative data presented in this figure can be found in S1 Data. Created in BioRender, https://BioRender.com/e01g235.

FRCaMPi for further characterization and named the selected variant SomaFRCaMPi (Fig 3A-3C). The RPL10 peptide was previously used to generate the soma-localized versions of the GCaMP6 sensors, which is a component of the 60S ribosome subunit [18]. To understand whether the SomaFRCaMPi is a competitive soma-targeting indicator in red, we generated fusions of K-GECO1, XCaMP-R, and jRGECO1a with the same tag and linker for comparison and referred to them as SomaK-GECO1, SomaXCaMP-R, and SomajRGECO1a (S4B Fig).

First, we characterized the subcellular localization of all generated constructs compared to the corresponding non-targeted GECIs in cultured mouse neurons. The ribo-tagged indicators exhibited a much steeper fluorescence decrease along the neurites compared to non-targeted versions. Specifically, ribo-tagged indicators have fluorescence dropped to 36.8% of soma value (we use $e^{-1}$-based function for fitting and quantification here, which is often used for characterizing the dropping speed of traits exhibiting an exponential decay) at a distance that was only 5-times shorter than that of the non-targeted version, indicating a strong elimination of expression from neurites by ribo-tagging (Fig 3D-3G). Previous studies have suggested that ribo-tagging could decrease the baseline fluorescence, resulting in dimmer constructs [22,25]. Similarly, we found that ribo-tagging resulted in a decreased baseline fluorescence of all tested red indicators in neuronal culture, with the extent differed for each indicator. Pairwise comparison of corresponding soma-targeted and non-targeted biosensors revealed that soma localization reduced brightness by 7.38 folds for SomajRGECO1a and by 4.85 folds for SomaXCaMP-R while SomaFRCaMPi and SomaK-GECO1 received much lower decreases in baseline (2.24 and 1.47 folds, respectively; S4D Fig). Consequently, SomaFRCaMPi demonstrated a baseline 2.64- and 2.16-fold brighter than that of SomajRGECO1a and SomaXCaMP-R, respectively (Fig 3H).

We then performed functional characterization of SomaFRCaMPi by adapting electrical field stimulation, the same as was applied to non-targeted indicators described above. Similar to the characterization of non-targeted indicators in neurons, SomaFRCaMPi exhibited the highest sensitivity (peak $\Delta F/F_0$ = 1.74 at 1 pulse) among all tested indicators (Fig 3I-3K). Furthermore, we found that the peak $\Delta F/F_0$ of SomaFRCaMPi preserved a degree of monotonical increase to the growing number of pulses similar to FRCaMPi with no obvious saturation detected up to 160 pulses (Fig 3I). The peak response of SomaFRCaMPi was more pronounced relative to non-targeted GECIs in comparison, showing the largest peak $\Delta F/F_0$ in all the numbers of stimuli applied and approximately 2-fold peak difference response to SomajRGECO1a at numbers of pulses in the range from 40 to 160 (Fig 3I and Table 2). Notably, despite the initial bleaching, the SNR of SomaFRCaMPi was still 90–180% higher compared to all other indicators at 1 pulse and stayed at least twice as high as that of SomajRGECO1a (Fig 3M and Table 2) across all assessed pulse numbers, in consonance with the large peak $\Delta F/F_0$ and a higher baseline observed for SomaFRCaMPi (Fig 3H and 3I). In comparison to the non-targeted indicators, SomaFRCaMPi also exhibited 22–112% higher peak $\Delta F/F_0$ relative to that of FRCaMPi between pulse numbers from 1 to 80 (Fig 3N). However, in other soma-localized indicators, we observed either a stronger decrease in baseline brightness or a significant drop in sensitivity such as for SomaK-GECO1 (S4E and S4H Fig). SomaFRCaMPi shares a similar half-rise kinetics to that of SomaXCaMP-R and SomaK-GECO1 (Fig 3O and Table 2) and showed a faster half-rise time than SomajRGECO1a. The decay kinetics of SomaFRCaMPi remained slower but showed a lesser degree of difference to that of other soma-localized indicators compared to the case in non-targeted GECIs (compare results in Figs 2H and 3P, and see Table 2 for the extracted numerical values). Those results demonstrate the high sensitivity and superior capability of SomaFRCaMPi to report calcium signals in neuronal culture. We thus excluded SomaXCaMP-R, SomaK-GECO1a, and their non-targeted versions from further characterization in vivo, referring to their much lower dynamic range and SNR in neuronal culture.

## Characterization of SomaFRCaMPi in zebrafish larvae

The transparent nature of zebrafish larvae [26] makes imaging of the entire fish brain possible with spatial precision and cellular resolution for studying the function of neuronal circuits involved in behaviors [27,28]. Similar to the application in mouse studies, RPL10 peptide has been used in zebrafish as a tool for cell type-specific transcriptomic profiling [29,30],

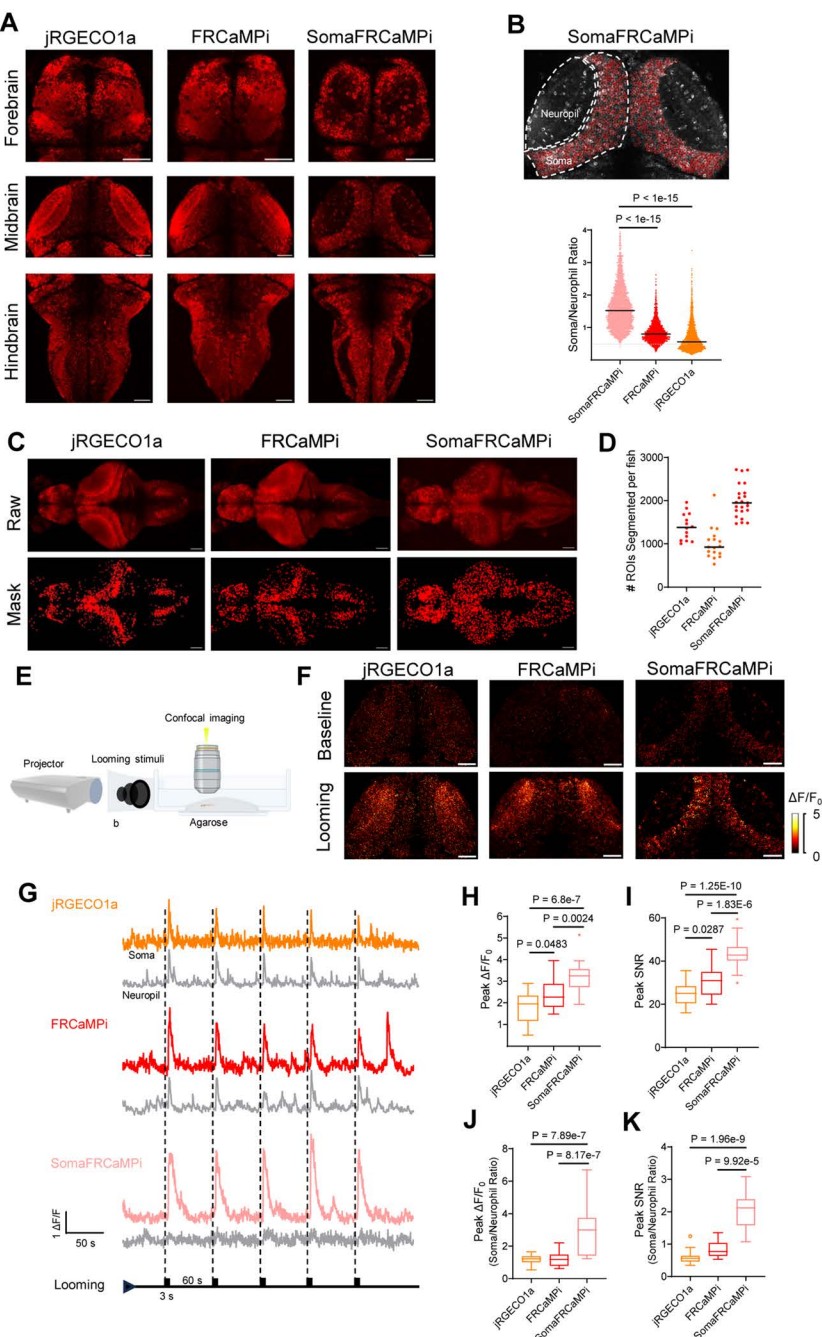

**Fig 4. Expression and performance of jRGECO1a, FRCaMPi, and SomaFRCaMPi in reporting neuronal activities in zebrafish larvae. (A)** Representative single plane confocal images showing fore-, mid-, and hindbrain regions of live zebrafish larva expressing jRGECO1a, FRCaMPi, and SomaFRCaMPi in neurons driven by the HuC promoter (n = 10 fish at 6 dpf for each). **(B)** Representative single-plane confocal image of optical tectum showing the segmentation of somatic and neuropil fluorescence (top) and plotted soma/neuropil ratio of fluorescence (bottom) (jRGECO1a: 2,597 cells from 6 fish; FRCaMPi: n = 1,826 cells from 6 fish; SomaFRCaMPi: n = 2,884 cells from 6 fish). **(C)** Representative max projection confocal images (top) and segmentation masks (bottom) by Cellpose (n = 14 fish for jRGECO1a, n = 16 fish for FRCaMPi, n = 23 fish for SomaFRCaMPi). **(D)** Number of ROIs detected per fish larva in Cellpose (n = 14 fish for jRGECO1a, n = 16 fish for FRCaMPi, n = 23 fish for SomaFRCaMPi). **(E)** Schematics of looming stimulation set up in which black expanding circular shadows projected in front of the larval zebrafish embedded in agarose gel. **(F)** Representative confocal images showing fluorescence response of optical tectum before and during looming stimulation periods (n = 3 fish each). **(G)** Representative single trial fluorescence traces of soma and neuropil region of red GECI-expressing neurons during looming stimulation shown as ticks (n = 18 cells from 3 fish

each). **(H–K)** Peak $\Delta F/F_0$ **(H)**, peak SNR at neuronal soma **(I)**, soma-to-neuropil peak $\Delta F/F_0$ ratio **(J)**, and soma-to-neuropil peak SNR ratio **(K)** of the fluorescence response during looming stimulation ($n$ = 18 cells from 3 fish each). One-way ANOVA test was used, followed by a post hoc Tukey test. The Shapiro–Wilk test was used for the normal distribution test. Scale bar, 50 μm. In dot plots, dots are individual data points; the line denotes the median. Box indicates the median and 25–75th percentile range, and the whiskers represent 1.5 times the interquartile range. The quantitative data presented in this figure can be found in S1 Data. Created in BioRender, https://BioRender.com/q72h354.

but has not yet tethered to reporting neuronal activity in vivo. We, therefore, investigated whether SomaFRCaMPi with ribo-tag motif could also be used in zebrafish for functional imaging, and if so, whether this would have any advantages over non-target sensors in fish. In this regard, we generated transgenic lines that pan-neuronally express jRGECO1a, FRCaMPi, and their soma-targeted variants SomajRGECO1a and SomaFRCaMPi followed by a side-by-side comparison of their performance in zebrafish larvae. However, during establishing stable lines, we observed that SomajRGECO1a in zebrafish was significantly dimmer compared to other indicators in the F1 generation. Specifically, the baseline brightness of SomajRGECO1a was still approximately three times lower than that of SomaFRCaMPi, despite the implementation of a 2.0-fold higher laser power for SomajRGECO1a (S7A Fig). Additionally, the fluorescence response of SomajRGECO1a upon stimulation was limited, despite acquired using a laser power (0.74 mW/mm$^2$) 10 times to that for the other three indicators (0.076 mW/mm$^2$; S7 Fig and S3 Movie). These factors complicated the selection of SomajRGECO1a-expressing zebrafish for breeding and the establishment of stable transgenic lines for sensors' assessment in F3 generation. Consequently, we decided to exclude SomajRGECO1a from further breeding efforts.

We first characterized the expression of indicators in larvae aged at 6 days post-fertilization (dpf) in F3 generation. The baseline brightness of the three indicators in fish larvae was comparable, with jRGECO1a being 28% and 17% brighter than FRCaMPi and SomaFRCaMPi, respectively (S8A Fig). We observed that in SomaFRCaMPi-expressing larvae, cell bodies could be more easily identified with minimal neuropil fluorescence found in the regions enriched with neuropils (Fig 4A) and in stratified neuropils in cell-body region (e.g., the tectal periventricular cells; S8B Fig). A quantification of soma-to-neuropil fluorescence in optical tectum further shows that fish larvae expressing SomaFRCaMPi exhibit a soma-to-neuropil fluorescence ratio more than twice to that of fish expressing non-targeted indicators, implicating a restriction of SomaFRCaMPi expression to neuronal soma (Fig 4B) in fish. With clearer cell bodies observed in SomaFRCaMPi-expressing larvae, we next investigated whether ribo-tagging facilitates the identification of neuronal soma in fish larvae by automated segmentation of maximum projection images using Cellpose [31]. Larvae expressing SomaFRCaMPi resolved hundreds to thousands more segmented regions of interest (ROIs), including the neuronal cell bodies obscured by neuropil structures such as those in the optical tectum, resulting in at least two times as high as those in larvae expressing FRCaMPi (Fig 4C and 4D).

To understand the dynamic performance of SomaFRCaMPi in zebrafish compared to non-targeted versions, we first examined fluorescence response in fish delivering external electrical stimulation pulse to larvae (S8C Fig), which evoke brain-wide and highly reproducible responses that facilitate benchmarking the sensors side-by-side (1 s duration, 60 V). Neuronal responses from cell bodies in superior medulla oblongata were recorded. We observed the highest peak $\Delta F/F_0$ and SNR attributed to SomaFRCaMPi, followed by FRCaMPi (S8E Fig). All three indicators possess a similar half-rise kinetics, while SomaFRCaMPi and FRCaMPi activities share a slower half decay kinetics compared to that of jRGECO1a (S8F Fig).

We then examined the feasibility of SomaFRCaMPi to report Ca$^{2+}$-induced fluorescence response in larvae, presenting the larvae with looming visual stimuli, which can be considered as more biologically relevant compared to external electrical shock (Fig 4E). Neuronal activity was recorded from optical tectum where neurons were found to be robustly activated due to inputs from retinal ganglion cells (RGC) during looming [32]. Looming induced clear and reproducible responses in neurons expressing SomaFRCaMPi, FRCaMPi, and jRGECO1a (Fig 4F and 4G). Similarly to the findings from external electrical stimulation, we found that tectal cell bodies expressing SomaFRCaMPi demonstrate the highest peak $\Delta F/F_0$ of

325% upon stimulation. More significantly, the peak SNR at cell bodies for SomaFRCaMPi larvae was 70% higher that of jRGECO1a. FRCaMPi larvae further showed 16% higher peak $\Delta F/F_0$ (Fig 4H) and 24% larger SNR (Fig 4I) than those of jRGECO1a. Notably, we observed a much weaker fluorescence response in tectal neuropil in fish expressing SomaFRCaMPi (Fig 4F and 4G) and, accordingly, much higher folds of soma-to-neuropil ratio in peak fluorescence response (Fig 4J) and SNR (Fig 4K) for SomaFRCaMPi fish. The pattern of both half rise and decay kinetics between the three indicators are similar to the observation in electrically evoked neuronal activities (S8G Fig). Together, those results demonstrate the high structural and functional compatibility of ribo-tagged FRCaMPi to zebrafish, highlighting the facilitation of reporting somatic neuronal activity in fish by SomaFRCaMPi and demonstrating the superior sensitivity of SomaFRCaMPi in fish.

**Two-photon calcium imaging of neuronal populations in mouse brain**

Two-photon (2P) microscopy offers exceptional optical sectioning and deep tissue imaging capabilities, enabling the acquisition of high-contrast, cell-resolution functional images [33,34]. This makes it particularly suitable for in vivo imaging of scattering tissues, such as the mouse brain. For in vivo mouse experiments, we utilized 2P imaging to evaluate the calcium dynamics of FRCaMPi and SomaFRCaMPi. Using AAVs, we expressed jRGECO1a, FRCaMPi or their soma-targeting variants into the primary visual cortex and recorded activities from L2/3 neurons of awake mice in response to drifting gratings presented in eight orientations (Fig 5A). Consistent with previous reports of 2P imaging of ribo-tagged GCaMPs [22,25], we first found that the fluorescence of SomaFRCaMPi was confined to neuronal cell bodies whereas the expression of FRCaMPi and jRGECO1a contributed to a much broader neuropil fluorescence surrounding (Fig 5B). Under identical experimental conditions, the expression of SomajRGECO1a resulted in many fluorescent foci with nuclear filling, and no fluorescence dynamics was detected during the recording of spontaneous activity ($n$ = 3 mice; Fig 5B). Therefore, we did not examine SomajRGECO1a further.

   The basal fluorescence of FRCaMPi in 2P imaging was 70% of that of jRGECO1a, while SomaFRCaMPi had approximately half the basal fluorescence of FRCaMPi (S9A Fig). This level of decreased baseline brightness in SomaFRCaMPi compared to FRCaMPi corresponded to that found in mouse neuronal culture (S4D Fig). Despite the twice lower basal fluorescence of SomaFRCaMPi, we used the same imaging parameter for fair comparison among all indicators to record visual stimulus-evoked activity. We found that the L2/3 neurons expressing all three indicators exhibited specific response selectivity to grating orientation (Fig 5C). For kinetics, we applied a sampling frequency (33 Hz) that was similarly used in GCaMP8 characterization [8] and observed 1.4-fold slower half-decay kinetics in FRCaMPi (0.32 s) compared to jRGECO1a (0.23 s). Strikingly, the half-decay time of SomaFRCaMPi (0.19 s) is similar to jRGECO1a and 41% faster than that of FRCaMPi (S9B ang S9C Fig). Meanwhile, the total number of responsive neurons extracted by SomaFRCaMPi from a single field of view (FOV) in L2/3 (930 neurons/16 FOVs = 58.125 neurons/FOV) was twice that obtained with FRCaMPi (30.45 neurons/FOV) and 2.6 times that achieved with jRGECO1a (21.9 neurons/FOV) from multiple mice. To further validate that the improvement in neuronal segmentation can be achieved in brain regions with different neuroanatomy, we imaged the parabrachial nucleus in the brainstem, where the neuronal population density is much higher than in the cortex (2.73 versus 0.94 neurons per gram of brain tissue) [35]. We conducted imaging across a large volume with a field of view of 0.8 × 1.0 mm², comprising four *z*-stacks with no overlapping neurons (S10A and S10B Fig). SomaFRCaMPi enabled automatic segmentation of twice as many neuronal ROIs per field of view compared to FRCaMPi (S10C Fig) and reported activity from thousands more neurons when datasets were combined (S10D Fig). These findings demonstrated that SomaFRCaMPi facilitates improved segmentation and allows for high-throughput recording of neuronal activities in two-photon imaging modalities from brain regions with varying neuronal density levels. We then quantified and compared the sensitivity of the three GECIs by adapting two established metrics, the proportion of responsive neurons detected and the cumulative distribution of peak $\Delta F/F_0$ from each neuron [4]. The fraction of responsive cells from recordings

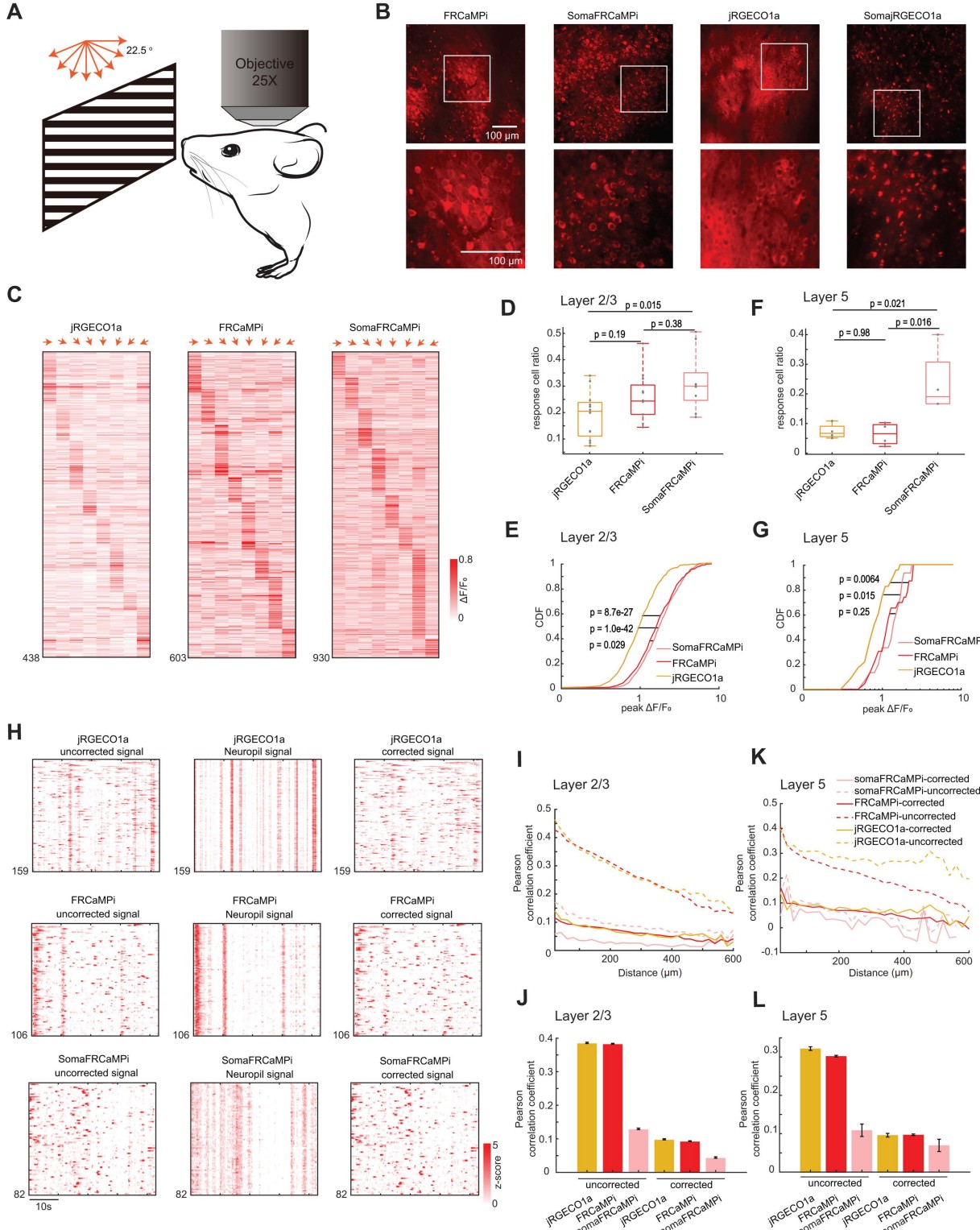

**Fig 5. In vivo neural population imaging in mice V1 cortex. (A)** Schematics of the imaging experiment with drifting grating visual stimulation. **(B)** Representative single-plane two-photon images of V1 neurons expressing FRCaMPi, SomaFRCaMPi, jRGECO1a, and SomajRGECO1a (FRCaMPi: $n$ = 20 FOVs from 5 mice; SomaFRCaMPi: $n$ = 16 FOVs from 4 mice; jRGECO1a: $n$ = 20 FOVs from 5 mice; SomajRGECO1a: $n$ = 14 FOVs from 5 mice). **(C)** Trial averaged response amplitudes of single responsive neurons in V1 for different angles of drifting gratings indicated by red arrows. **(D)** Box

plot of fraction of recorded V1 neurons responding to moving grating stimuli (jRGECO1a, *n* = 12 FOVs from 5 mice; FRCaMPi, *n* = 12 FOVs from 5 mice; SomaFRCaMPi, *n* = 9 FOVs from 3 mice) **(E)**. Cumulative distribution function of neural response ($\Delta F/F_0$) to the preferred stimuli for V1 layer 2/3 (L2/3) neurons (jRGECO1a: *n* = 438 cells from 5 mice; FRCaMPi: *n* = 609 cells from 5 mice; SomaFRCaMPi: *n* = 930 cells from 3 mouse). Two-sample Kolmogorov–Smirnov test was used. Central line inside the box, median; box, interquartile range; whiskers, maximum and minimum; outliers, individual data points. One-way ANOVA test was used followed by Tukey's multiple comparison. **(F and G)**. Same as **(D)** and **(E)** but for V1 layer 5 (L5) neurons. *n* = 23 cells, 4 FOVs from 2 mice (jRGECO1a), 23 cells, 4 FOVs from 2 mice (FRCaMPi), and 15 cells, 4 FOVs from 2 mice (SomaFRCaMPi). Two-sample Kolmogorov–Smirnov test was used. **(H)** Spontaneous calcium activity of V1 neurons expressing jRGECO1a, FRCaMPi, and SomaFRCaMPi from representative sessions. Calcium traces of either uncorrected (left), neuropil signal (middle), or corrected with surrounding neuropil signal subtraction (right). **(I)** Correlation coefficients of spontaneous calcium activities between pairs of V1 L2/3 neurons plotted against the centroid distance (jRGECO1a: *n* = 82,765 neuron pairs from 4 mice; FRCaMPi: *n* = 310,392 neuron pairs from 4 mice; SomaFRCaMPi: *n* = 50,355 neuron pairs from 4 mice). **(J)** Correlation coefficients of spontaneous calcium activities between pairs of V1 L2/3 neurons that are within 100 μm for data shown in **(H)**. **(K and L)** Same as **(H)** and **(I)**, but for V1 L5 neuron pairs (jRGECO1a: *n* = 24,220 neuron pairs from 3 mice; FRCaMPi: *n* = 114,782 neuron pairs from 3 mice; SomaFRCaMPi: *n* = 1,721 neuron pairs from 3 mice). The quantitative data presented in this figure can be found in S1 Data.

of the FRCaMPi- and SomaFRCaMPi-expressing mice was 1.21 to 1.52-fold higher than those of jRGECO1a (Fig 5D). In parallel, the cumulative distribution of peak $\Delta F/F_0$ of FRCaMPi and SomaFRCaMPi was substantially shifted over larger values, indicating FRCaMPi and SomaFRCaMPi being more sensitive than jRGECO1a (Fig 5E). The median peak $\Delta F/F_0$ amplitude for FRCaMPi and SomaFRCaMPi was more than 177% to that of jRGECO1a. Consecutively, the fraction of responsive cells of SomaFRCaMPi is higher than FRCaMPi (median 29 and 23%, respectively; Fig 5D) and we also observed a slightly right-shifted distribution of peak $\Delta F/F_0$ of SomaFRCaMPi relative to that of FRCaMPi (median 1.75 and 1.60, respectively; Fig 5E). The observed higher sensitivity level for SomaFRCaMPi during 2P-imaging remained consistent with that observed in neuronal cultures and zebrafish. In addition, FRCaMPi and SomaFRCaMPi also showed higher-value distribution of SNR in L2/3 neurons compared to jRGECO1a (S9D Fig). Notably, the distribution of SNR values for SomaFRCaMPi (median 7.59) is only 7% less to that of FRCaMPi (median 8.17, *p* = 0.08) despite of a lowered baseline brightness by half.

To further assess the dynamics and sensitivity of our red indicators during deep tissue imaging in vivo, we recorded visual stimulus-evoked activities from L5 neurons (Fig 5F and 5G). We observed a very similar pattern in the peak $\Delta F/F_0$ distribution for the three indicators to those seen in L2/3 (Fig 5G). The median peak $\Delta F/F_0$ amplitudes for FRCaMPi and SomaFRCaMPi remained 43–68% higher than that of jRGECO1a. Notably, the fraction of responsive cells in recordings from mice expressing SomaFRCaMPi (median 19%) was almost three times that of FRCaMPi and jRGECO1a (medians 6.6% and 6.7%, respectively; Fig 5F). Meanwhile, in L5 neurons, we found that the SNR of jRGECO1a, FRCaMPi, and SomaFRCaMPi were converging. Nonetheless, FRCaMPi exhibited the highest SNR, followed by SomaFRCaMPi and jRGECO1a (medians 6.57, 6.43, and 6.35, respectively; S9E Fig), similar to observations in L2/3. These results demonstrate that the indicators are capable of reporting neuronal activities from deeper cortical layers under two-photon (2P) imaging, albeit with lower sensitivity compared to 2P imaging in L2/3 neurons.

Given the outstanding peak $\Delta F/F_0$ and peak SNR of SomaFRCaMPi found in visual cortex, we extended our investigation to its long-term expression performance, specifically analyzing Ca²⁺ spikes from spontaneous activity recordings. In response to prevalent concerns in the field regarding nuclear filling [2] and reduced indicator responses during prolonged expression [2,13], we sought to address these issues with SomaFRCaMPi. Notably, neither SomaFRCaMPi nor jRGECO1a exhibited significant nuclear filling after 4 months of expression in V1 L2/3, with only minimal nuclear filling observed (S9H Fig). At 6 weeks of expression, SomaFRCaMPi demonstrated 46% higher peak $\Delta F/F_0$ compared to jRGECO1a and a similar SNR in spontaneous spikes (S9I-S9K Fig). In the same mice after 4 months of expression, both indicators demonstrated a drop of sensitivity with around a 30% drop of peak $\Delta F/F_0$ (S9I and S9J Fig) compared to that of 6 weeks. However, the decline in peak SNR for SomaFRCaMPi was only 4.5% (median: 8.87 at 6 weeks versus 8.48 at 4 months; *p* = 8.9e–27), approximately half the reduction observed in jRGECO1a (−8.7%, median: 8.87 at 6 weeks versus 8.48 at 4 months; *p* = 4.5e–38) (S9K Fig). Consequently, even after 4 months of expression, SomaFRCaMPi maintained 38% higher peak $\Delta F/F_0$ and a slightly higher SNR compared to jRGECO1a. These results suggest that SomaFRCaMPi

can reliably report neuronal activity over extended periods with less degradation in SNR, highlighting its potential for stable calcium imaging in the long term.

In conjunction with findings from the study on ribo-tagged GCaMPs [22,25], we found highly synchronized neuropil fluorescence in recordings of FRCaMPi- and jRGECO1a-expressing mice but not in that for SomaFRCaMPi (Fig 5H). To understand the extent to which SomaFRCaMPi reduces neuropil contamination in 2P imaging, we analyzed distance-wise correlation of activities between neurons from the recording of spontaneous events in V1. Using raw imaging datasets for analysis, we initially observed a correlation level of around 0.45 in activity between neurons expressing FRCaMPi (mean 0.43) and jRGECO1a (mean 0.46), which decreased to 0.13 (same for FRCaMPi and jRGECO1a) with increasing distance to 600 µm. This correlation was 2- to 7-fold lower in neurons expressing SomaFRCaMPi compared to both FRCaMPi and jRGECO1a (Fig 5I). We then implemented a standard algorithmic approach to subtract neuropil signals from neuronal traces as previously suggested [22,36]. This adjustment resulted in a more than a 3-fold reduction in signal correlation between neurons expressing either jRGECO1a or FRCaMPi. We further quantified correlation coefficient of neuron pairs within 100 µm (Fig 5J). Post-subtraction, the signal correlation levels of jRGECO1a (mean 0.099) and FRCaMPi (mean 0.093) were slightly lower than those of uncorrected SomaFRCaMPi (mean 0.13) but remained within a very similar degree of low correlation. Neurons expressing SomaFRCaMPi also exhibited a slight drop in correlation following signal correction. The mean correlation coefficient of corrected signals was still more than 50% lower than those of jRGECO1a and FRCaMPi after correction. Moreover, the pattern of comparison in correlation was found similar in neuronal activity from L5 (Fig 5K and 5L). Therefore, SomaFRCaMPi substantially reduces the neuropil-induced correlation between cells in 2P imaging experiments with superior sensitivity levels, demonstrating its capacity as a competent soma-localized GECI in red for 2P imaging.

To further evaluate the sensitivity of SomaFRCaMPi compared to top-performing green sensors in the field, we co-expressed RiboL1-GCaMP8s and SomaFRCaMPi in the nucleus of the solitary tract (NTS) of the brainstem. RiboL1-GCaMP8s was chosen as the most sensitive soma-GECIs reported to date as it has been found to exhibit a similar fluorescence response in vivo to non-targeted jGCaMP8s, one of the most sensitive green calcium indicators to date (see Fig S5 in the reference [25]). Neuronal activity was examined using dual-color 2P calcium imaging in response to multiple internal organ stimuli (Fig 6A and 6B). Neurons co-expressing green and red calcium probes showed selective and dose-dependent functional responses to increasing gastric distension, characterized by a growing number of responding neurons and elevated fluorescence response amplitudes (Fig 6C and 6D), validating the functionality of the NTS calcium imaging system. Next, we compared the peak responses from NTS neurons that co-expressed both indicators. We found that the overall peak ΔF/F level of SomaFRCaMPi was approximately 33% less than that of RiboL1-GCaMP8s, as demonstrated by least-squares fitting (Fig 6E). However, we observed an overall lower noise in the red channel (Fig 6D). Specifically, examination of the baseline noise level in SomaFRCaMPi showed it to be approximately 32% lower than that of RiboL1-GCaMP8s (Fig 6F), resulting in an almost identical Z-score between the two probes (Fig 6G). These results indicate that SomaFRCaMPi exhibits lower imaging noise and comparable sensitivity to a state-of-the-art green soma-GECI.

## Wide-field primary neocortical imaging of awake mice using SomaFRCaMPi

Provided with the faithful neuropil elimination and the enhanced sensitivity of SomaFRCaMPi under 2P microscopy, we next investigated the benefits of using SomaFRCaMPi in awake behaving mice using one-photon widefield microscopy, which is a more accessible and less complex system compared to 2P setups. Challenges in neuron detection and deteriorated somatic signals [37] were often observed in one-photon widefield or micro-endoscopic imaging of the mouse brain due to tissue scattering and the strong out-of-focus background fluorescence [38,39]. Restricting the expression of indicators to neuronal somas, in turn, may help to directly enhance the clarity of somatic calcium signaling and improve signal accuracy in wide-field imaging [23] (S1 Text). So far, optical recording with single-cell resolution using either one-photon micro-endoscopic or widefield imaging modality was dominated by green GECIs [40–45], while few studies have reported

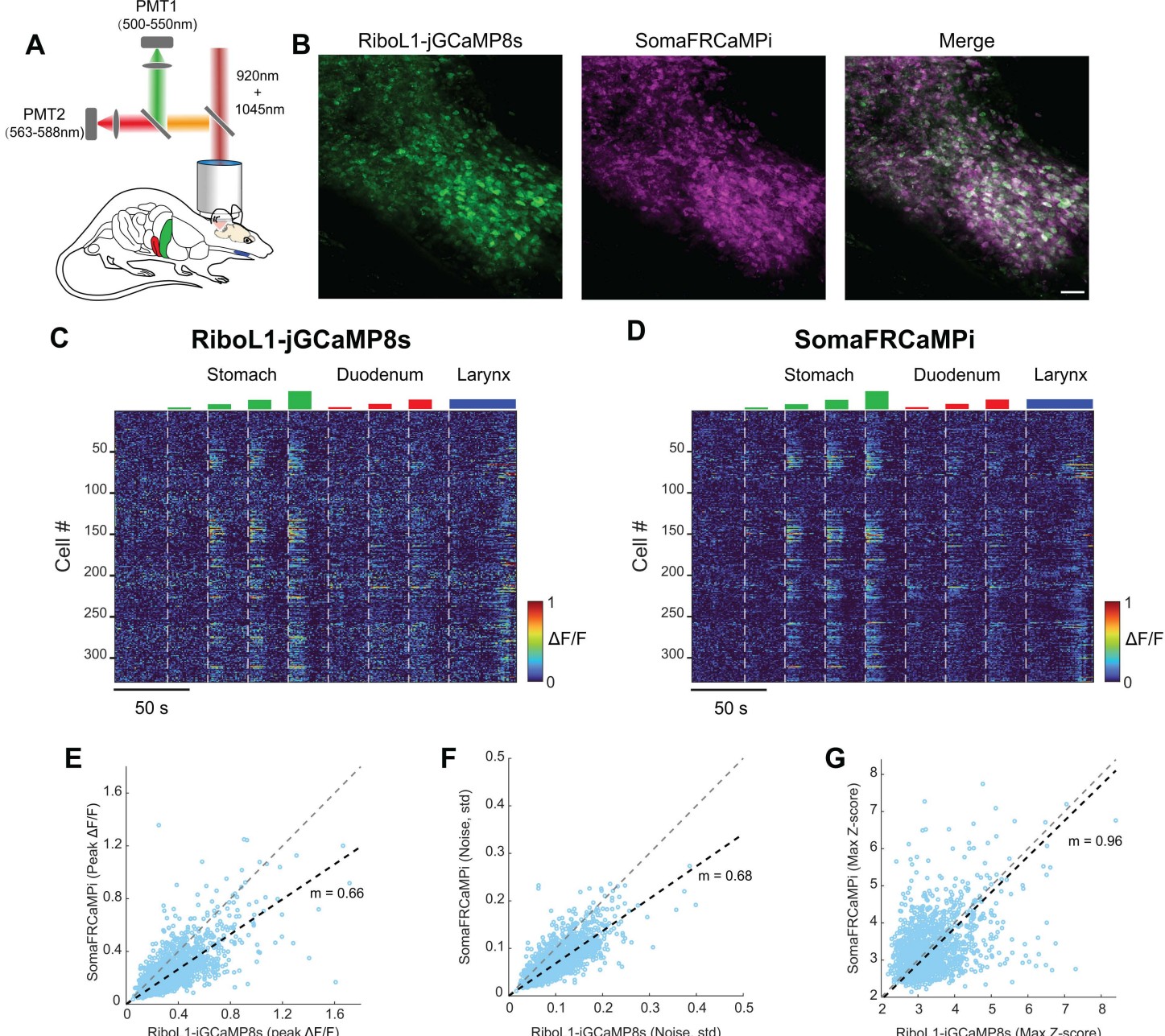

**Fig 6. Dual-color in vivo calcium imaging of NTS tuning during gastric distension. (A)** Schematics depicting NTS two-photon calcium imaging setup during gastric distension. **(B)** Representative two-photon NTS images showing the co-expressing of RiboL1-jGCaMP8s and SomaFRCaMPi with the shared ROIs identified from Cellpose segmentation. Scale bar, 50 μm. **(C and D)** Heat maps depicting time-resolved responses to a series of gastric distensions from NTS neurons co-expressing Ribo-GCaMP8s **(C)** and SomaFRCaMPi **(D)** from the image in **(B)** (329 neurons). **(E)** Scatter plot showing the peak ΔF/F of green (Ribo-GCaMP8s) and red (SomaFRCaMPi) calcium responses from NTS neurons co-expressing both indicators during gastric distension. **(F–G)** Similar to **(E)**, but for noise evaluation **(F)** and peak Z-score analysis **(G)**. $n$ = 1,580 neurons from 2 mice. The grey diagonal dash represents a slope equal to 1, and the black dashed line denotes the linear fit of the data. The quantitative data presented in this figure can be found in S1 Data.

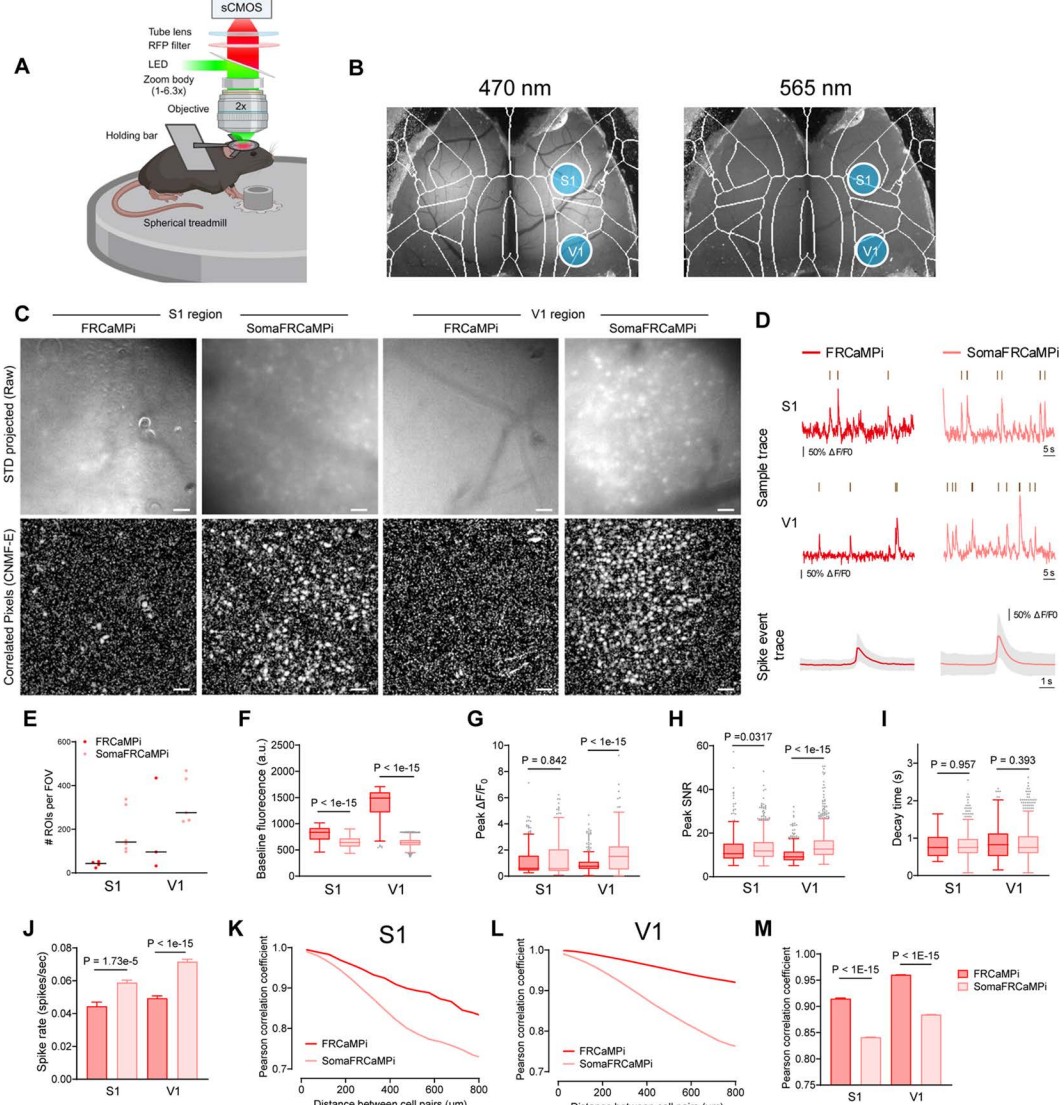

**Fig 7. Recording of FRCaMPi and SomaFRCaMPi dynamics in S1 and V1 using wide-field microscopy. (A)** Schematics of the wide-field one photon imaging experiment in awake resting mice. **(B)** Representative macroscopic images showing pan-cortical expression of gGRAB$_{5-HT3.0}$ excited by 470 nm (left) and SomaFRCaMPi excited by 565 nm (right) (*n* = 4 mice). **(C)** Representative images (the standard deviation of images from one imaging session) for the S1 or V1 region in either SomaFRCaMPi- or FRCaMPi-expressing mice (top) and the corresponding correlation image by CNMF-E (bottom) (*n* = 4 mice each). **(D)** Top, representative calcium fluorescence traces with calcium events identified for FRCaMPi and SomaFRCaMPi from the experiments shown in **(C)** (*n* = 249 neurons from 5 FRCaMPi mice and *n* = 1,081 neurons from 5 SomaFRCaMPi mice). Bottom, aligned calcium events displayed as mean and S.D. **(E)** Number of ROIs detected per FOV by CNMF-E (S1: *n* = 4 FOVs from 4 FRCaMPi mice, *n* = 6 FOVs from SomaFRCaMPi 4 mice; V1: *n* = 3 FOVs from 3 FRCaMPi mice and *n* = 5 FOVs from 5 SomaFRCaMPi mice). In dot plots, line denotes median. **(F–H)** Baseline brightness **(F)**, peak $\Delta F/F_0$ **(G)**, and peak SNR **(H)** of SomaFRCaMPi compared to FRCaMPi in S1 and V1 (S1: *n* = 161 neurons from 4 FRCaMPi mice, *n* = 1,143 neurons from 4 SomaFRCaMPi mice. V1: *n* = 563 neurons from 3 FRCaMPi mice, *n* = 1,653 neurons from 5 SomaFRCaMPi mice). **(I)** The decay time of SomaFRCaMPi compared to FRCaMPi in S1 and V1 (S1: *n* = 56 neurons from 4 FRCaMPi mice, *n* = 595 neurons from 4 SomaFRCaMPi mice. V1: *n* = 343 neurons from 3 FRCaMPi mice, *n* = 1,247 neurons from 5 SomaFRCaMPi mice). **(J)** Spike rate of SomaFRCaMPi compared to FRCaMPi in S1 and V1 (S1: *n* = 37 neurons from 4 FRCaMPi mice, *n* = 364 neurons from 4 SomaFRCaMPi mice. V1: *n* = 212 neurons from 3 FRCaMPi mice, *n* = 717 neurons from 5 SomaFRCaMPi mice). **(K)** Pearson correlation coefficient of calcium dynamics plotted against distance for different cell pairs from S1 region of mice expressing FRCaMPi (*n* = 3,410 cell pairs from 5 mice) or SomaFRCaMPi (*n* = 136,576 cell pairs from 4 mice). **(L)** As in **K** but for V1 (*n* = 99,451 cell pairs from 3 FRCaMPi mice and *n* = 296,784 cell pairs from 5 SomaFRCaMPi mice). **(M)** Pearson correlation coefficient of calcium dynamics between pairs of neurons within 800 μm. Statistics are the same as in **K** and **L**. Mann–Whitney U test. Scale bar, 50 μm. Bar chart is shown as mean ± s.e.m. Box indicates the median and 25–75th percentile range, and the whiskers represent 1.5 times the interquartile range. Spike rate and Pearson correlation coefficient are plotted as bars to aid the demonstration of data distribution. See S6 Table for more details. The quantitative data presented in this figure can be found in S1 Data. Created in BioRender,.https://BioRender.com/t42r557.

on the extraction of single-neuron activity using red GECIs using wide-field imaging [46]. Meanwhile, those studies were often facilitated by achieving optimal neuronal labeling density [40–42,44–46] and/or special optics such as holographic illumination patterns [43,46] to reduce background and improve cell signal detection (S1 Text). Provided with the superior sensitivity of our red probes in zebrafish and mice, we, therefore, investigated whether SomaFRCaMPi could report population neuronal activity with single-cell resolution and improved accuracy in mouse cortex under a widefield imaging setup.

For this experiment, we pan-cortically expressed either SomaFRCaMPi or FRCaMPi in the mouse cortex (S11A and S11B Fig) and installed a polymer cranial window that spans the dorsal cortex of mice [47]. We then head-fixed the mouse and imaged neuronal activity using a wide-field stereoscope [48] (Fig 7A). By tuning the stereoscope to the highest spatial resolution achievable for our setup (zoom body set to 6.3×), we achieved a millimeter-sized FOV for populational imaging of neurons. A green indicator, GRAB-5HT3.0, was co-expressed to help assure pan-neuronal expression pattern and enhance visualization of the vasculature morphology as a reference for locating specific brain regions at higher magnifications (Fig 7B and S2 Movie). We chose to image the primary somatosensory (S1) and visual cortex (V1), which are considered to be two of the densest cortical regions, with 110,000 ± 1,000 and 164,000 ± 50,000 neurons per mm³, respectively [49]. Thus, obtaining quality somatic signals would be, in principle, more challenging in those regions compared to others due to a stronger labeling density in the background. To ensure a fair comparison between non-targeted and soma-localized versions of FRCaMPi, and to evaluate the contribution of neuropil contamination to distance-dependent neuronal activity correlation, neuronal activity was recorded only while the mouse was resting on the treadmill. This approach was taken to prevent additional input from correlated neuronal activities in the somatosensory and visual cortex that could arise due to locomotion, as suggested by previous studies [50–52]. As a result, we successfully detected $Ca^{2+}$ spikes from neuronal soma in both S1 and V1 of brains expressing FRCaMPi and SomaFRCaMPi (Fig 7C and 7D). We identified many more spiking neurons by simple visual assessment of the raw datasets (S5 Movie). In parallel, we observed a consistently high labeling density of 80–90% of cortical neurons estimated at the imaging site using post hoc histological analysis (S11C Fig). Further analysis of the number of ROIs segmented by CNMF-E [38] confirmed that the number of ROIs identified per FOV for SomaFRCaMPi in S1 and V1 was 3.3 and 2.9 times larger than that for FRCaMPi, respectively (Fig 7E). Consistent with observation in mouse neuronal culture and in mouse in vivo under 2P, the baseline fluorescence of SomaFRCaMPi neurons in mouse cortex was about 23% and 57% dimmer than that of FRCaMPi ones under the same imaging parameters in S1 and V1, respectively (Fig 7F). The profiling of neuronal spikes showed similar peak $\Delta F/F_0$ between FRCaMPi- and SomaFRCaMPi-expressing mice in S1 and 2-fold peak $\Delta F/F_0$ for SomaFRCaMPi spikes in V1 (Fig 7G), consistent with the observation in 2P mouse imaging. However, the peak SNR of SomaFRCaMPi spikes from both brain regions were higher than FRCaMPi ones, with 10% and 40% higher peak SNR observed for SomaFRCaMPi spikes in S1 and V1, respectively (Fig 7H). This indicates that the expression of SomaFRCaMPi reduces the overall background noise to a level that achieves a higher SNR despite owning a lower brightness. The decay kinetics remained very similar between SomaFRCaMPi and FRCaMPi obtained from individual spike waveforms (Fig 7I). Meanwhile, SomaFRCaMPi neurons also reported approximately 50% more calcium events than those of FRCaMPi in FOV from both brain regions (Fig 7J).

We next measured the distance-wise correlation of neuronal activity from neuron pairs within the FOV of Soma-FRCaMPi versus FRCaMPi mice. To achieve this, we analyzed the Pearson correlation coefficient (PCC) of raw data traces extracted from ROIs as previously suggested [22,23]. In S1 and V1 of mice expressing either of the indicators, we observed correlation coefficients of 0.73–0.99 between cell pairs, with the drop of correlation found for cell pairs at increasing distances over the range of 0–800 μm. The magnitude of correlation was much higher than that observed in 2P imaging experiments, and the correlation difference between FRCaMPi and SomaFRCaMPi was also much higher. As suggested by Chen and colleagues [22], optical scattering and out-of-focus fluorescence likely contribute to the overall blurriness observed in wide-field imaging and, thus, dominate high correlation by noise. Despite presenting a much denser spiking population and a higher spiking rate on top (Fig 7C, 7E and 7J), SomaFRCaMPi still exhibited a lower correlation

coefficient across the broad range of distances with a steeper decrease of coefficient between neuronal pairs, ending up at a level approximately 20% to 30% less than FRCaMPi at 800 μm (Fig 7K-M). We further analyzed the mean correlation coefficient for distance ranges: 0–200, 200–400, and 400–800 μm and found correlations at different distance ranges were all significantly lower in SomaFRCaMPi-expressing mice compared to FRCaMPi ones (S11D Fig). To summarize, in both S1 and V1 regions, SomaFRCaMPi reduced contamination to a less signal correlation, enabled detection of more neurons with a higher number of neuronal spikes, and increased SNR using the widefield setup. These results collectively underscore the efficacy of SomaFRCaMPi for one-photon widefield imaging, demonstrating its advanced ability as a red soma-localized indicator to report more somatic signals with higher precision in this noisy imaging modality.

## Discussion

Advancements in microscopic techniques [39,40,53,54] and image processing algorithms [55–59] have profoundly improved the proficiency of in vivo calcium imaging to achieve concurrent profiling of large populational neuronal activities with cellular resolution, making it possible to address computational and coding principles of functional brain units beyond single neurons [39,60]. This advancement can be further enhanced by the optimization of GECIs. For example, improved sensitivity and soma restriction for green GECIs were shown to improve somatic signal detection [22,23,25], decrease neuropil contamination [22,23,25], and reduce erroneous spikes [23] in calcium imaging in vivo. In this study, we reported FRCaMPi and its soma-localized version, SomaFRCaMPi, a pair of red calcium indicators with high dynamic range and excellent sensitivity. Furthermore, we highlighted the advantages of using SomaFRCaMPi for signal extraction and precision recording of populational neuronal activities in mice and zebrafish in vivo.

It is advantageous to use GECIs with affinity that match the resting $Ca^{2+}$ level of the cellular system, as the maximal slope of the fluorescence change in their $Ca^{2+}$ titration curve occurs near the resting $Ca^{2+}$ concentration (S1D Fig), maximizing their detectability of changes in $[Ca^{2+}]$ [6]. In neuronal calcium imaging, tuning affinity of GECIs closer to resting intracellular $[Ca^{2+}]$ of neurons (i.e., approximately 30–100 nM) [61,62] could contribute to improved detection amplitude of changes (i.e., $\Delta F/F_0$) in $Ca^{2+}$ in neuronal activities (e.g., the lower $K_d$ and higher $\Delta F/F_0$ of the sensitive variant in each generation of GCaMPs reported) [2,3,8]. Previous strategies for improving the sensitivity of GECIs often include random or semi-rational mutagenesis and functional screening of selected candidates [2,3,8,9,16,17]. A recent study on ultra-high affinity GECIs based on single fluorescent protein (1-FP) may provide an alternative way to generate sensitive mutants [21]. By inverting topology and further optimizing linker length between CaM and the CaM binding peptide [63], $K_d$ of GECIs were tuned down to less than 30 nM (5–8 folds lower than their precursor). Those indicators enabled the detection of fluorescence change in amoebae that possess an ultra-low resting $[Ca^{2+}]$ (approximately 10 nM) and small intracellular dynamics (<100 nM) [21], which is not accessible to most GECIs designed for sensing neuronal calcium dynamics. In this study, similarly, we found that by a simple topological inversion of FRCaMP, the sensitivity of the indicator was increased in detecting neuronal calcium dynamics (S2B Fig). The $K_d$ value of FRCaMPi (81 nM) is much closer to the resting intracellular $[Ca^{2+}]$ in neurons compared to its precursor (214 nM), which likely explains the improved sensitivity in neuronal recording. This approach offers a shortcut to enhance indicator sensitivity without the need for laborious mutant screening.

To find an effective soma-targeting motif for engineering a red soma-localized GECI, we conducted a peptide screening and chose the RPL10 tag to generate SomaFRCaMPi for characterization in neuronal culture and in vivo (S6 Fig). While soma-localization improved soma restriction and functional imaging in GCaMPs, it could significantly hamper the functionality of the probe such as baseline brightness [22,23,25]. To mitigate this issue, the fusion of RPL10 to FRCaMPi avoids direct linkage of the structurally dynamic calcium-interactive part of the sensor, the M13 peptide, to localization peptide that could anchor the probe rigidly to ribosomes in cells. By systematical characterization in cultured neurons, the resulting indicator, SomaFRCaMPi, indeed outperformed the three other ribo-tagged red GECIs that are all based on cpRFP. Out of all ribo-tagged red indicators examined, SomaFRCaMPi showed both the highest sensitivity with peak $\Delta F/F_0$ of approximately 180% per single AP and only 2-fold lower brightness compared to the non-targeted version (S4D

Fig). A substantial decrease of baseline fluorescence was seen in the case of SomaXCaMP-R (about 5 folds) and Soma-jRGECO1a (>7 folds), and a significant drop in fluorescence responses was observed in SomaK-GECO1 (> 2.3 folds; S4 Fig). Given that current red GECIs exhibited significantly lower brightness and weaker fluorescence responses compared to optimized green GECIs [13], a further substantial reduction of brightness and/or response amplitude could significantly influence their performance in vivo. This likely explains the absence of reports on soma-localized red GECIs to date. Another successful example showing the benefit of the ncp design in fusions is the generation of a bioluminescent GECI, LUCI-GECO1, based on a genetic fusion of NanoLuc luciferase domain to ncpGCaMP6s, showing a 2.7-fold higher bio-luminescent resonance energy transfer (BRET) efficiency than fusion to the original GCaMP6s [20,24]. Together, we think the inversion topology for the existing GECIs might be more compatible with genetic fusions for preserved functionality of the indicator and may facilitate subcellular targeting to other organelles.

For in vivo applications, we demonstrated both the structural and functional compatibility of RPL10 fusion used for calcium imaging in fish larvae. Structurally, SomaFRCaMPi reduced neuropil fluorescence substantially in zebrafish and enabled segmentation of a doubled number of neuronal somas compared to non-targeted FRCaMPi. Functionally, Soma-FRCaMPi demonstrated exceptional sensitivity in zebrafish. Notably, in the looming experiment, SomaFRCaMPi has shown the ability to clearly differentiate neuropil signals from somatic signals in the optic tectum. Given the advancements in zebrafish imaging technologies and data analyses [27,64–66], implementing ribo-tagged indicators would allow for the extraction of a greater number of neurons with improved signal precision, advancing the outcome of whole-brain imaging. More strikingly, applying ribo-tagging to FRCaMPi did not compromise its basal fluorescence in zebrafish larvae (S8A Fig), a phenomenon whose underlying reasons remain unclear given the variability of expression observed across different bio-logical systems [22]. Therefore, integrating those advanced properties of SomaFRCaMPi holds promise for more precise, sensitive and comprehensive neural imaging in zebrafish studies.

For imaging in mice, we utilized both 1P and 2P microscopy to elucidate the advantages of using SomaFRCaMPi in functional recording of neuronal populations. SomaFRCaMPi enhanced signal detection with improved SNR in densely labeled neuronal populations during wide-field 1P imaging. It was also sufficient to reliably report activity in deeper corti-cal layers under 2P imaging and improved recording precision by reducing artificial signal correlations in both 1P and 2P imaging. These observations highlight the critical features of soma-localized indicators, consistent with those observed in soma-localized GCaMPs [22,23], underscoring the effectiveness of using SomaFRCaMPi in mice. Previous reports have shown that jRGECO1a exhibits a sensitivity level comparable to that of GCaMP6f but performs 40% worse than GCaMP6s in L2/3 of V1 in vivo [13]. In our study, we extended this comparison by evaluating SomaFRCaMPi against the latest generation of GCaMP indicators. Side-by-side characterization of RiboL1-GCaMP8s and SomaFRCaMPi fluores-cence responses in neurons of the NTS demonstrated comparable Z-score levels, along with a 30% reduction in baseline noise for SomaFRCaMPi. These results highlight the advantages of using a red fluorescent probe for imaging in dense tissue such as brain stem and confirm the outstanding sensitivity of SomaFRCaMPi, matching that of top-performing soma-targeted GECIs.

Intriguingly, SomaFRCaMPi demonstrated a higher sensitivity level compared to the non-targeted FRCaMPi, both in vitro and in vivo. The enhanced sensitivity resulting from ribo-tagging likely stems from several factors. Firstly, sensitivity may be directly influenced by the targeted sensor design, as evidenced by the varied effects on sensitivity observed for the other three ribo-tagged red GECIs based on cpRFPs (S4G Fig). Secondly, fusions to different localization peptides using the same indicator might lead to differential changes in $\Delta F/F_0$, as seen in the peptide screening results of this study and in the study by Shemesh and colleagues [23]. Another possible explanation for the increased sensitivity in Soma-FRCaMPi is the resultant $Ca^{2+}$ buffering in neurons. In contrast to non-targeted FRCaMPi localized throughout the entire neuron, including neuropils, the total $Ca^{2+}$ buffering capacity of SomaFRCaMPi is presumably weaker due to fewer sensor molecules present in the cytoplasm since SomaFRCaMPi expression was restricted to neuronal soma. Consequently, as compared to FRCaMPi, upon calcium influx, the SomaFRCaMPi sensor molecules have access to more $Ca^{2+}$, resulting

in a stronger fluorescence change and, thus, a higher $\Delta F/F_0$. This reduced buffering should also accelerate the kinetics of the indicator, as the reduced resistance to $Ca^{2+}$ influx would make both the kinetics of $Ca^{2+}$ influx and the indicator more reflective of their true dynamics. Although we did not observe a significant improvement in kinetics in zebrafish for Soma-FRCaMPi to support the hypothesis of reduced $Ca^{2+}$ buffering, we did observe faster kinetics for SomaFRCaMPi versus FRCaMPi in neuronal culture and under 2P imaging in mice evoked by visual stimuli (Figs 2, 3 and S9C).

In this study, we showcased the advantages of combining topological inversion and soma restriction to re-engineer an mApple-based red calcium indicator for higher sensitivity and improved performance in population calcium imaging. The resulting indicator, SomaFRCaMPi, achieved applications and image quality similar to those of soma-targeted green GECIs. However, the limitations of the current study should be considered. Since our goal was to demonstrate the application of topological inversion using FRCaMP as an example, the amino acid sequence of both FP and calcium-sensing domain is not perturbed. As a result, FRCaMPi retains certain characteristics of FRCaMP, such as suboptimal photostability and off-kinetics. We also anticipate that photoactivation of mApple-based sensors may be a concern for combing FRCaMPi with blue-light driven optogenetic tools. Since FRCaMPi was engineered from the mApple derivative of R-GECO1, it is likely to exhibit a similar degree of photoswitching as R-GECO1. In addition, the topological inversion enhanced calcium binding affinity, which led to slower calcium dissociation and further prolonged the off-kinetics of the indicator as a trade-off (S2C Fig). Both FRCaMP and FRCaMPi are based on mApple RFP, which could lead to complications such as fluorescent aggregates (S11A and S11B Fig). Nevertheless, those should indicate a great opportunity and space to optimize the indicator further for better applications. Recent studies hold promise in eliminating undesired features in existing red GECIs [67,68] and engineering red GECI using better performing RFP than mApple is likely to lead to GECIs with improved properties.

In addition, topological inversion can be explored in alternative ways, presenting valuable opportunities for sensor optimization. By analogy, the engineering of a topological variant can also be reversed. For instance, inverting the topology of a 1-FP potassium indicator, GINKO1 ($K_d$ = 2.38 mM), which originally had a non-circular permuted (ncp) topology, resulted in cp-GINKO1 with a substantially reduced ligand binding affinity ($K_d$ = 270 mM) [21]. Consequently, the pursuit of topological variants may be a more general strategy, as changes in ligand affinity are necessary to optimize signal detection in biological or subcellular targets with either low or high ligand concentrations. This strategy of reversing topological configurations (between cp and ncp) could also be applicable to sensors other than GECIs, as were shown for potassium (ncp to cp) and glutamate (cp to ncp).

In conclusion, we have developed a pair of new red GECIs by integrating altered topological structures and fusing them with a soma-localization motif. Unlike traditional screening methods, our approaches rapidly yielded indicators with improved peak $\Delta F/F_0$, demonstrating markedly higher sensitivity compared to one of the leading competitors, jRGECO1a. The high sensitivity against jRGECO1a was validated across all tested biological imaging modalities, with minimal compromise to other properties of the indicators. Our results suggest that utilizing topological inversion can be a valuable strategy for re-engineering 1-FP-based GECIs, thereby enhancing $Ca^{2+}$ detectability. Furthermore, additional genetic fusions using the ncp topology design should be encouraged, as this approach holds promise for diverse applications of 1-FP GECIs. While FRCaMPi already exhibited excellent dynamic range and superior sensitivity, SomaFRCaMPi should be of particular interest as another color choice for spatial precision and large-population recording of neuronal activities in mammalian cells and in vivo.

## Materials and methods

### Molecular cloning

The FRCaMPi gene was generated using PCR with overlapping fragments [69] using Fw-LSSmOrange-BglII, RSP-EcoRI-r, CaMSp-M13, CaMSp-M13-r, and RSPi-EcoRI-r primers listed in S2 Table. The FRCaMPi gene was cloned into

the pBAD/HisB-TorA-mTagBFP plasmid at BglII/EcoRI restriction sites using the Fw-LSSmOrange-BglII/RSPi-EcoRI-r primers to express FRCaMPi variant in the periplasm of bacterial cells. To construct the pAAV-*CAG*-NES-FRCaMPi plasmid, the FRCaMPi gene was PCR-amplified as the BglII-EcoRI fragment using the Fw-LSSmOrange-BglII/RSPi-EcoRI-r primers and swapped with the mCherry gene in the pAAV-*CAG*-NES-mCherry vector.

Synthetic DNA oligonucleotides used for cloning were synthesized by Tsingke Biotechnology. PrimeStar Max master mix (Takara) or 2×Hieff PCR Master Mix (Yeasen) was used for high-fidelity polymerase chain reaction (PCR) amplifications. Restriction endonucleases were purchased from New England BioLabs and used according to the manufacturer's protocols. DNA ligations were performed using T4 DNA ligase (New England BioLabs) or the NovoRec plus one step cloning kit (Novoprotein). The ligation products were chemically transformed into the TOP10 *E. coli* strain (Biomed) and cultured according to the standard protocols. Sequencing of bacterial colonies and purified plasmids was performed using Sanger sequencing (Zhejiang Youkang Biological Technology).

Small-scale isolation of plasmid DNA was performed with commercially available Mini-Prep kits (Qiagen or Tiangen). The genes of red GECIs were de novo synthesized by Tsingke Biotechnology or Shanghai Generay Biotech based on the cDNA sequences reported in the original publications. Oligonucleotides for cloning were purchased from Tsingke Biotechnology or Zhejiang Youkang Biological Technology (S2 Table). To clone pAAV-CAG-NES-FRCaMPi-P2A-mGreenLantern, pAAV-CAG-XCaMP-R-P2A-mGreenLantern and pAAV-CAG-NES-jRGECO1a-P2A-mGreenLantern the complementary DNA of mGreenLantern was first PCR amplified with SpeI/EcoRI flanking sites and swapped with the EGFP gene in the pAAV-CAG-miRFP670–2-P2A-GFP plasmid (WeKwikGene plasmid #0000007). Then, the cDNA of NES-FRCaMPi, NES-jRGECO1a or XCaMP-R was PCR amplified with KpnI/ AgeI flanking sites and swapped with the miRFP670−2 gene in the pAAV-CAG-miRFP670–2-P2A-GFP. To clone pAAV-CAG-NES-K-GECO1-P2A-mGreenLantern and the cDNA of K-GECO1 was PCR amplified with BglII/ AgeI flanking site and swapped with the FRCaMPi gene in the pAAV-CAG-NES-FRCaMPi-P2A-mGreenLantern plasmid. We fused nuclear export sequence (NES) to the N-termini of red GECIs except for XCAMP-R where the F2A sequence appears to be an NES. To clone pAAV-Syn-NES-Red-GECI, the cDNA of NES-jRGECO1a, NES-KGECO1 or XCaMP-R was PCR amplified with BamHI/EcoRI flanking site and swapped with the CoChR-GFP gene in the pAAV-Syn–CoChR–GFP (Addgene plasmid #59,070). From the peptide screening result, we choose to use ribo-tag to engineering a soma-localized version of FRCaMPi strictly localized to soma. Originally inspired by the SomaGCaMP paper [23], we aimed to facilitate protein folding by adding a flexible linker (GGSGGSGGT)3 from SomaGCaMP6f2 to the C-terminus of FRCaMPi. SomaFRCaMPi was then generated by linking RPL10. To clone pAAV-Syn-NES-FRCaMPi-RPL10, the cDNA fragment of NES-FRCaMPi and RPL10 was PCR amplified with overlapping fragments using Syn-NES-F, FRCaMPi-R, Linker27-F and WPRE- Linker27-R primers in S2 Table. To clone pAAV-Syn-NES-K-GECO1-RPL10, pAAV-Syn-XCaMP-R-RPL10 and pAAV-Syn-NES-jRGECO1a-RPL10, the cDNA fragment of NES-K-GECO1, XCaMP-R or NES-jRGECO1a and RPL10 was PCR amplified with overlapping fragments using Syn-NES-F, Syn-XCaMPR-F, Syn-NES-jRGECO1a-F, Linker27-XCaMPR-R, Linker27-jRGECO1a-R, Linker27-K-GECO1-R and WPRE-Linker27-R primers in S2 Table.

## Protein purification and characterization

Proteins were expressed and purified as described in [70]. All experiments for protein characterization were performed at ambient temperature of 21–23°C.

The extinction coefficient values for the FRCaMPi indicator in a $Ca^{2+}$-saturated state or apo-state were calculated in 30 mM HEPES buffer, pH 7.2 (adjusted by KOH), 100 mM KCl (buffer A), supplemented with 5 mM $CaCl_2$ or 10 mM EDTA, respectively, using the alkaline denaturation method and assuming that the mCherry-like red chromophore had an extinction coefficient of 38,000 $M^{-1}$ $cm^{-1}$ at 455 nm in 1 M NaOH [14]. The absorption spectra were registered using a NanoDrop 2000c Spectrophotometer (Thermo Scientific, DE, USA)

To determine the quantum yield of FRCaMPi excited at 540 nm in a $Ca^{2+}$ saturated state or apo-state, the integrated fluorescence values (in the range of 550–750 nm) were measured in buffer A supplemented with 5 mM $CaCl_2$ or 10 mM EDTA, respectively, and compared with the same values for the equally absorbing at 540 nm mCherry protein (quantum yield of 0.22 (ref. [14])). To determine the quantum yield of the purified FRCaMPi protein excited at 400 nm in the apo-state, the integrated fluorescence values (in the range of 410–750 nm) were measured in buffer A supplemented with 10 mM EDTA and compared with the same values for the mTagBFP2 protein equally absorbing at 400 nm (quantum yield of 0.64 (ref. [71])). The fluorescence spectra were acquired using a CM2,203 spectrofluorometer (Solar, Minsk, Belarus).

pH titrations for FRCaMPi (50 nM final concentration) in a $Ca^{2+}$-saturated state or apo-state were performed in buffer: 30 mM citric acid, 30 mM borax, and 30 mM NaCl with pH values ranging from 3.0 to 11.0 (adjusted by HCl or NaOH), supplemented with 0.1 mM $CaCl_2$ or 0.1 mM EDTA, respectively, followed by incubation for 20 min, as described in [70]. Red fluorescence (Ex: 525 nm/Em: 580–640 nm) was registered using a 96-well ModulusTM II Microplate Reader (Turner Biosystems, Sunnyvale, CA, USA).

To determine the equilibrium calcium $K_d$, the fluorescence of the FRCaMPi protein (50 nM final concentration) was measured in a mixture of two stock buffers: 30 mM 3-(N-morpholino) propanesulfonic acid (MOPS), pH 7.2 (adjusted by KOH), 100 mM KCl supplemented with 10 mM EGTA or 10 mM Ca-EGTA (buffer C), as described previously [70]. Buffer B and C were mixed to 0, 0.25, 0.5, 1, 2, 3, 4, 5, 6, 7, 8, 9, 9.5, 9.75, and 9, 875, and 10 mM $Ca_{total}$ concentrations to obtain 2.1, 7.9, 16.3, 35.4, 79.2, 135.5, 208.4, 312.6, 468.9, 731.5, 1254.6, 2817.6, 5856.0, 11858.0, 21923.7, and 81,276 nM $Ca_{free}$ concentrations (the free calcium concentrations were corrected by 1.084-fold according to the GCaMP6s indicator $K_d$ at 144 nM (ref. [2])). To determine the equilibrium calcium $K_d$ in the presence of 1 mM $Mg^{2+}$ ions, the fluorescence of the FRCaMP protein (50 nM final concentration) was measured in a mixture of buffer B and C supplemented with 1 mM $MgCl_2$. The dilutions of the B and C buffers solutions were the same, and free calcium concentrations were considered as described above in the absence of $Mg^{2+}$ ions. Red fluorescence (Ex: 525 nm/Em: 580–640 nm) was registered using a 96-well ModulusTM II Microplate Reader (Turner Biosystems, Sunnyvale, CA, USA).

Photobleaching experiments were performed with suspensions of purified proteins in mineral oil, as previously described [72]. Briefly, the photobleaching were measured using purified proteins dialyzed in buffer A supplemented with either 10 mM EDTA or 5 mM $CaCl_2$ at a 50 µM concentration in aqueous microdroplets in mineral oil using a Zeiss Axio Imager Z2 microscope (Zeiss, Germany) equipped with a 120 W mercury short-arc lamp (LEJ, Germany), a 63 × 1.4 NA oil immersion objective lens (PlanApo, Zeiss, Germany), a 550/25 BP excitation filter, a FT 570 beam splitter, and 605/70 BP emission filters for red channel and 470/40 BP excitation filter, a FT 495 beam splitter, and 525/50 BP emission filters for green channel. Light power densities (red channel 9 mW/cm$^2$ and green channel 16 mW/cm$^2$) were measured at the rear focal plane of the objective lens using a PM100D power meter (ThorLabs, Germany) equipped with an S120VS sensor (ThorLabs, Germany). No corrections were applied to the experimental data.

## Characterization in HeLa cell culture

HeLa (ATCC CCL-2) cells were cultured in DMEM supplemented with 10% fetal bovine serum (FBS) and 1% penicillin–streptomycin, and seeded in a 24-well glass-bottom plate (P24-0-N Cellvis) after Matrigel (356,235, BD Biosciences) coating. Cells at 80–90% confluency were transfected with 500 ng of DNA (pAAV-CAG-NES-RedGECI-P2A-mGreenLantern) using Hieff Trans Liposomal Transfection Reagent (Yeasen Biotechnology, 40802ES02) and cultured for. Before imaging, the medium was replaced with 0.5 mL $Ca^{2+}$ and $Mg^{2+}$ free Hanks balanced salt solution (HBSS). To quantify the brightness level of red GECIs in HeLa cells, images were acquired in both TRITC (excitation 561 nm at 1.53 mW/mm$^2$) and FITC (excitation 488 nm at 2.5 mW/mm$^2$) using a ×20 NA 0.75 objective (Nikon). The mean fluorescence intensity in the TRITC and FITC channels for ROIs were extracted and the red-to-green fluorescence ratios were calculated after background subtraction for each channel. To obtain photobleaching curve of red GECIs in HeLa cells, images were acquired using the TRITC channel at a laser power of 7.4 mW/mm$^2$.

To record histamine-induced $Ca^{2+}$ oscillation of red GECIs, 5 mM histamine and 1 M $CaCl_2$ was added to create a final concentration of 50 µM and 6 mM of $CaCl_2$. After 10 min of imaging, 1 µl 2.5 mM ionomycin was added to make a final 5 µM ionomycin solution and cells were imaged for another 2–6 min to capture ionomycin-induced calcium influx. Image was acquired using TRITC (1.53 mW/mm$^2$) with a ×20 NA 0.75 objective (Nikon). All images were acquired using Nikon Eclipse Ti inverted microscope, SPECTRA X light engine (Lumencor), and OrcaFlash4.0v2 camera (Hamamatsu), controlled by NIS-Elements AR software. The maximal dynamic response, $\Delta F_{max}/F_{min}$, was defined as maximal fluorescence change during the period of ionomycin-induced calcium influx, where $F_{min}$ is the minimal fluorescence value of a single frame before fluorescence increase and $\Delta F_{max} = F_{max} - F_{min}$, where $F_{max}$ is the maximal fluorescence value of a single frame during ionomycin-induced response. Data were analyzed offline using NIS Elements Advance Research software (versions 5.21.00 and 5.30.00), Excel (Microsoft), and GraphPad Prism 9.

## Characterization in primary neuronal culture

In S2 Fig, dissociated neuronal cultures were isolated from the C57BL/6 mice at postnatal days 0–1 and were grown on a 24-well cell imaging black plate with a glass bottom coated with 0.001% polylisine; then, the tissue cultures were treated (Eppendorf, Germany) in Neurobasal Medium A (GIBCO, UK) supplemented with 2% B27 Supplement (GIBCO, UK), 0.5 mM glutamine (GIBCO, UK), 50 U/ml penicillin, and 50 µg/ml streptomycin (GIBCO, UK). The rAAV particles for neuronal transduction were purified from 10 to 150 cm dishes, as described in the original paper [73]. On the 4th day in vitro, neuronal cultures were transduced with a mixture of rAAV viral particles carrying AAV-CAG-NES-FRCaMPi and AAV-CAG-NES-NCaMP7. The cells were imaged using an Andor XDi Technology Revolution multi-point confocal system on DIV 16 (spontaneous activity at 37 °C, 5% carbon dioxide) and DIV 17–18 (electrical field stimulation at 21–23 °C).

Stimulation of neuronal cultures was performed using a self-built electrical system described earlier [73]. In this step, 300 voltage pulses of a 1 ms duration (0.5 ms negative phase, 0.5 ms interphase, and 0.5 ms positive phase) at an 87 Hz frequency with an amplitude of ±70 V were applied to the neuronal cultures in 24-well plates through iridium electrodes with a 5 mm gap. 10 µM cyanquixaline (6-cyano-7-nitroquinoxaline-2,3-dione) (CNQX) and 100 µM (2R)-amino-5-phosphonovaleric acid (APV or AP5) were added before stimulation to block spontaneous neuronal activity. To quantify the fluorescence intensity, the background noise determined from the adjacent cell-free area was subtracted from mean fluorescence intensity value for the cytosolic sub-region of the cell of the similar area. The rise and decay half-times were calculated as time difference between time point corresponding to the calcium spike maximum and time points at half-maximum at the left and right edges of the spike, respectively.

For functional profiling of cytosolic-expressing and soma-localized red GECIs (Figs 1, 2 and S4), cultured neurons were transduced with viruses at 5 DIV. Constructs used for transduction included: AAV2/9-Syn-K-GECO1-WPRE, AAV2/9-Syn-SomaK-GECO1-WPRE, AAV2/9-Syn-XCaMP-R-WPRE, AAV2/9-Syn-SomaXCaMP-R-WPRE, AAV2/9-Syn-jRGECO1a-WPRE, AAV2/9-Syn-SomaFRCaMPi-WPRE (Shanghai Sunbio Medical Biotechnology). All data presented were obtained from transduced neurons, except for those in Figs 1C, 2F, S4A and S4B, neurons were co-transfected with pAAV-Syn-Red-GECI-WPRE and pAAV-CAG-emiRFP670-P2A-GFP (WeKwikGene #0000006) by the calcium phosphate transfection method as used previously. To compare different red sensors side-by-side in neuronal culture, the volume of each AAV was diluted and adjusted to approximately $1 \times 10^{13}$ vg/ml of AAV particles using titer number provided by the manufacture. All the viruses used were not undergone number of free-thaw cycles more than twice. To obtain a high density of transduced neurons, approximately 1 µl of diluted viruses were applied per well of the 24-well plate. On DIV 11–13, neurons were transferred to a stimulation chamber (Warner Instruments, RC-49MFSH), stimulated and imaged in extracellular buffer solution containing 150 mM NaCl, 5 mM KCl, 10 mM glucose, 10 mM HEPES, pH 7.4, 2 mM $CaCl_2$, 1 mM $MgCl_2$. Electrical pulses (83 Hz) of 40–50 V, 1-ms width were evoked by field stimulation with the stimulation chamber and a voltage amplifier (ATA2041, Aigtek). A total number of eight trains containing increased numbers of electrical stimuli (1, 2, 3, 5, 10, 20, 40, 80, 160) were evoked for each image of neuronal trace. Images were acquired at 20 Hz using a

Nikon Eclipse Ti inverted microscope equipped with a ×20,0.75 NA objective (Nikon), TRITC (Ex: 561 nm, 1.53 mW/mm$^2$) from SPECTRA X light engine (Lumencor), and OrcaFlash4.0v2 camera (Hamamatsu), controlled by NIS-Elements AR software.

For electrical stimulation experiments in Figs 1, 2 and S4, neurons from all wells were recorded for 5 min before stimulation to remove any potential bleaching effect. ROIs corresponding to the soma of responsive neurons was manually selected from each train of a field stimulation trace. In neuronal culture, peak $\Delta F/F_0$ is defined as maximal fluorescence change of a spike train over $F_0$, where $F_0$ is the average of 1 s fluorescence baseline right before the stimulus. SNR is calculated as the maximal fluorescence change over the standard deviation of 1 s fluorescence baseline before the stimulus, reported as relative ratio. Neuronal ROIs for calculating peak $\Delta F/F_0$ and SNR were selected independently for different analytical experiment. In particular, ROIs for peak $\Delta F/F_0$ were selected from individual neurons with a single background whereas ROIs for peak SNR were selected from neurons with each paired to an adjacent background. To correct variation of fluorescence response observed in independent transduced cultures, the mean of peak $\Delta F/F_0$ and SNR between those cultures were normalized. Fluorescence trains from ROI was excluded from dataset when fluorescence spikes failed recovery to baseline or were not evoked by electrical stimulation.

To obtain the kinetics of the indicator, epochs from $\Delta F/F_0$ traces containing rise and decay stage of the trace were first manually extracted and fitted using a single exponential function using MATLAB to calculate their half kinetics. Half-rise ($t_{1/2,\ rise}$) and decay ($t_{1/2,\ decay}$) time were defined as the time to the half of peak $\Delta F/F_0$ in the rise phase and the time from peak $\Delta F/F_0$ to half of peak $\Delta F/F_0$ in the decay phases, respectively, calculated by interpolation. Neuronal ROI was excluded when fluorescence spikes failed recovery to baseline or has a fitting coefficient less than 0.8. For neuronal brightness of red GECIs, only neuronal soma was used to extract mean fluorescence intensity of the ROI. For the bleaching analysis, neurons that exhibited fluorescence spikes and did not return to baseline levels by the end of the imaging session were excluded. Data were analyzed offline using NIS Elements Advance Research software (versions 5.21.00 and 5.30.00), MATLAB (version 2022a), Excel (Microsoft), and GraphPad Prism 9.

## Animals

All animal maintenance and experimental procedures were conducted according to institutional and ethical guidelines for animal welfare, and all animal studies were approved by the Institutional Animal Care and Use Committee (IACUC) of Westlake University, Hangzhou (animal protocol #KP-19–044) or Tsinghua University (animal protocol #15-GZC3.G23-1), Beijing, China, or the Institutional Animal Care and Use Committee at The Scripps Research Institute or the National Research Center "Kurchatov Institute" Committee on Animal Care (NG-1/109PR of 13 February 2020) and were done in accordance with the Russian Federation Order Requirements N 267 M3 and the National Institutes of Health Guide for the Care and Use of Laboratory Animals. For all experiments involving mice throughout, the C57BL/6J strain (supplied by the animal facility of Westlake University or the laboratory animal research center at Tsinghua University or the Scripps Research Institute or the National Research Center "Kurchatov Institute") was used regardless of sex.

## Immunohistochemistry

Mice were deeply anesthetized with 1% pentobarbital sodium (10 μl/g) and perfused transcardially with phosphate-buffered saline (PBS) followed by 4% paraformaldehyde (PFA) in PBS. Brains were kept in PBS containing 4% PFA overnight. Coronal slices (50 μm thick) were cut using a vibratome (VT1200 S, Leica Microsystems) and then incubated in a staining solution composed of 3% (v/v) molecular Nissl stain (Invitrogen, N21483) and 0.1% (v/v) DAPI in PBS for 15 min. The slices were then rinsed in PBS, briefly dried, and mounted on the glass slides using prolong diamond antifade (Invitrogen, P36961, USA). For quantification of labeling density, after in vivo experimentation, imaging site was marked on the surface of polymer window using fast green. After transcardial perfusion, the tissue corresponding to the imaging site was cut and marked using a fine surgical scalpel. At least two slices were imaged from each imaged brain region and for

each indicator. Images were acquired using CSU-W1 SoRa imaging setup of Nikon Spinning-Disk Field Scanning Confocal System with 405, 561 and 647 nm excitation using a ×10 NA0.45 and ×20,0.80 objective lens (Nikon), controlled by NIS-Elements AR software.

For labeling density analysis, we first applied CLAHE [74] function in ImageJ to all channels to homogenize fluorescence intensity distribution and thereby to correct for possible bias in automatic segmentation due to locally dim or bright fluorescence. Then Cellpose [31] was used for segmentation with identical settings. The ROIs at the edge of image were removed. The labeling density was defined as the ratio between GECI-expressing and Nissl +ve ROIs.

## Confocal imaging of zebrafish larvae

**Zebrafish.** To generate transgenic zebrafish line, plasmid containing either pTol-HuC:NES-SomaFRCaMPi (25 ng/μL) or pTol2-HuC:NES-FRCaMPi (25 ng/μL) and Tol2 mRNA (25 ng/μL) were co-injected into fertilized eggs of background strain albino (slc45a2$^{b4}$), and the founders were screened at adult stage. Adult fish and larvae were maintained in a 14–10 h light-dark cycle at 28 °C. Adult zebrafish were raised and crossed with wild-type zebrafish until positive F3. Larval zebrafish do not undergo sex differentiation until 1-month fertilization. All the experiments were carried out using F3 larval zebrafish.

**In vivo confocal imaging.** All experiments were performed on 6 days-post-fertilization (dpf) larvae in 10% Hank's solution containing (in mM): 140 NaCl, 5.4 KCl, 0.25 $Na_2HPO_4$, 0.44 $KH_2PO_4$, 1.3 $CaCl_2$, 1.0 $MgSO_4$, and 4.2 $NaHCO_3$ (pH 7.2). Imaging of Tg (HuC:NES-FRCaMPi-NBL), Tg (HuC:NES-FRCaMPi) and Tg (HuC:NES-jRGECO1a) larvae at 5-dpf was performed with an FV3000 inverted confocal microscope (Olympus) by using a ×20 water-immersion objective (0.95 NA, morphology imaging). Larval zebrafish were mounted dorsal side up in 2.0% low melting-point agarose (Sigma–Aldrich) and then immersed in 10% Hank's solution. To image the sensor expression pattern, images were acquired using 561 nm (0.076 mW/mm$^2$) with a field of view of 1,024 × 1,024 pixels and spatial resolution of 0.622 × 0.622 × 5 μm$^3$ ($x × y × z$). Laser power of 0.74 mW/mm$^2$ was used to image F1 SomajRGECO1a and 0.37 mW/mm$^2$ was used to image F1 SomaFRCaMPi. For looming stimulation, black expanding dots on a blue background were chosen in case of spectral interference. The visual stimulation was given for 3 s with approximately 60 s interval. These images were acquired using 561 nm (0.076 mW/mm$^2$) with a field of view of 512 × 512 pixels with spatial resolution of 1.243 μm/pixel at approximately 3 Hz. Laser power of 0.74 mW/mm$^2$ was used to image F1 SomajRGECO1a.

For electrical stimulation experiment, field electrical stimuli (1 s, 60V) were then triggered by an Arduino board (Uno) with custom-written programs and delivered by platinum sheet electrodes and a voltage amplifier (*Iso-Flex*; AMPI). Imaging condition was the same as for looming stimulation experiment with sampling frequency of 4 Hz and 0.74 mW/mm$^2$ for 561 nm laser. We excluded those fish that failed to respond to the electrical stimulation.

**Analysis of confocal imaging of the zebrafish larvae.** For zebrafish time-lapse imaging in looming-evoked calcium activities, neuronal ROIs were manually selected according to morphology and Fiji was used to obtain the fluorescence intensity in the ROI in each frame. Peak $\Delta F/F_0$ is defined as maximal fluorescence change over $F_0$ of a looming-induced spike, where $F_0$ is defined as the 20th percentile value of fluorescence intensity within the sliding time window of 200 frames (about 65 s) which is 1/6 of the entire frames of recording. Peak SNR is calculated as $\Delta F_{max}/Std_{baseline}$, where $Std_{baseline}$ is the standard deviation of fluorescence intensity in the 20th percentile of the entire recording session. For kinetics analysis, epochs from $\Delta F/F_0$ traces containing rise and decay stage of the trace were extracted and fitted to a single exponential for calculating half-rise and decay time. Data analysis was done by custom-written MATLAB scripts. For quantifying number of neuronal ROIs per fish expressing GECI, Cellpose [31] was used for segmentation.

For analysis of evoked neuronal activity by electrical stimulation, a circular ROI with a diameter of 38.5 μm were selected in the superior medulla oblongata region for each fish. Peak responses and kinetic analysis were performed same as for looming. Statistical analyses were performed using GraphPad Prism 10.1.2.

## Fiber photometry recording

Stereotactic injections and fiber photometry experiments were performed as previously described [75]. For stereotactic surgeries, 10-week male mice were deeply anesthetized with isoflurane and kept under anesthesia during surgery. 200 nL of AAV2/9-Syn-FRCaMPi-WPRE was injected bilaterally in the central amygdala (CeA; AP: −1.3, ML: ±2.4, DV: 4.5 mm) followed by implantation of optical fiber (fiber core diameter: 200 μm; fiber length: 5.0 mm; NA: 0.37; Inper). Three months after virus expression, mice were adapted to head fixation for 30 min for consecutive 2 days. Water was restricted 24 h in head-restrained mice before training, and mice were trained to lick for water for 2 days. Then, Bpod was used to record behavioral events, and Inper fiber photometry was used to detect FRCaMPi signals in response to water and air puffs. The intensity of violet light ($\lambda = 410$ nm) was set at 10–20 μW and yellow light ($\lambda = 561$ nm) was 20–30 μW. Traces were extracted and data was analyzed by Inper Data Analysis.

## Two-photon calcium imaging of mouse parabrachial nucleus (PBN) neurons

**Stereotactic surgeries.** During surgery, mice were anesthetized with isoflurane (1.5–2%; flow rate, 1 L/min). Injections (approximately 250 nl total) were targeted to the PBN (AP: −5.15 mm; DV: −3.8 to −3.53 mm; ML: ±1.55 mm) to allow for the spread of the virus via efflux. For two-photon imaging of acute brain slices, we injected 250 nL of either a 1:1 mixture of AAV9-NES-SomaFR-CaMPi-WPRE and phosphate-buffered saline or a 1:1 mixture of AAV9-NES-FRCaMPi-WPRE and phosphate-buffered saline.

**Two-photon Imaging of acute brain slices.** All mice used for slice experiments were fed ad libitum. Approximately four weeks after viral injections, acute brain slices were prepared by deeply anesthetizing animals with isoflurane followed by rapid decapitation. Upon removal, brains were immediately immersed in ice-cold, carbogen-saturated (95% $O_2$, 5% $CO_2$) choline-based cutting solution consisting of (in mM): 92 choline chloride, 10 HEPES, 2.5 KCl, 1.25 $NaH_2PO_4$, 30 $NaHCO_3$, 25 glucose, 10 $MgSO_4$, 0.5 $CaCl_2$, 2 thiourea, 5 sodium ascorbate, 3 sodium pyruvate, oxygenated with 95% $O_2$/5% $CO_2$, measured osmolarity 310–320 mOsm/L, pH = 7.4. Then, 275−300 mm-thick coronal sections containing the PBN complex were cut with a vibratome (Campden 7000smz-2) and incubated in an oxygenated cutting solution at 34 °C for 10 min. Next, slices were transferred to oxygenated ACSF (126 mM NaCl, 21.4 mM $NaHCO_3$, 2.5 mM KCl, 1.2 mM $NaH_2PO_4$, 1.2 mM $MgCl_2$, 2.4 mM $CaCl_2$, 10 mM glucose) at 34 °C for an additional 15 min. Slices were then kept at room temperature (20–24 °C) for 45 min until use.

A single slice was placed in the recording chamber where it was continuously perfused with oxygenated ACSF at a rate of 2–5 mL per min at room temperature. Two-photon imaging was performed using a two-photon resonant-galvo scanning microscope (NeuroLabWare) controlled by Scanbox (https://scanbox.org/). An InSight X3 laser (Spectra-Physics) was used to excite the fluorophores (1,100 nm), and the emission light was filtered (Green: 510/84 nm; Red: 607/70 nm; Semrock) before collection with the photomultiplier tubes (PMT) (H10770B-40; Hamamatsu). The XY scanning was performed using resonant/galvomirrors and the Z scanning was achieved with an electrically tunable lens (Optotune). Imaging included 3–4 non-overlapping $z$-planes simultaneously in each slice. Imaging was performed with a 10 × 0.6 water-immersion Olympus objective. The excitation wavelength used was 1,100 nm. The field of view size was typically 0.8 × 1.0 mm$^2$.

**Analysis of two-photon imaging of PBN neurons.** Image registration for all two-photon calcium imaging was performed using Suite2p. For extraction of signals from cell body ROIs, we used Cellpose, specifically a custom model trained with equal numbers of Soma-FRCaMPi or FRCaMPi images (flow threshold = 0.5, cellprob threshold = −0.5). Fluorescence time series were extracted from the movement-corrected videos by averaging each of the pixels within each binarized mask. We calculated neuropil activity as the median value of an annulus surrounding each ROI (inner radius: 5 pixels; outer radius: 15 pixels; pixels belonging to any other ROI were excluded from these annulus masks). The time course of neuropil activity was subtracted from the activity time course of the associated ROI to create a fluorescence time course, $F(t)$, where $t$ is the time of each imaging frame. The change in fluorescence was calculated by subtracting a running estimate (10th percentile of a 60-second sliding window) of baseline fluorescence ($F_0(t)$) from $F(t)$, then dividing by $F_0(t)$: $dF/F(t) = (F(t) − F_0(t))/F_0(t)$.

## Two-photon Ca²⁺ imaging of mouse V1 cortical neurons

**Mouse surgeries for cortical imaging.** Adult C57BL/6J (Jax 000664) mice at 8–10 postnatal weeks were used for virus injection. During surgery, mice were anesthetized with isoflurane (1.5–2%; flow rate, 0.5–0.7 L/min). Flunixin meglumine (Sichuan Dingjian Animal Medicine Co., Ltd) was injected subcutaneously (1.25 mg/kg) during and after the surgery for at least three days to reduce inflammation. 50 nl of either AAV2/9-Syn-NES-FRCaMPi-RPL10-WPRE ($1.56 \times 10^{13}$ vg/mL), AAV2/9-Syn-FRCaMPi-WPRE ($1.25 \times 10^{13}$ vg/mL), AAV2/9-Syn-NES-jRGECO1a-RPL10-WPRE ($1.09 \times 10^{13}$ vg/mL) or AAV2/9-Syn-jRGECO1a-WPRE ($1.25 \times 10^{13}$ vg/mL) was injected in the primary visual cortex (AP: −3.0, ML: 2.5, DV: 0.25 mm). Two weeks after injection, mice were ready for craniotomy. A 3.0-mm-diameter craniotomy above injection site was made with a skull drill. After removing the skull piece, a coverslip was implanted on the craniotomy region, and a titanium headpost was then cemented to the skull for head fixation.

**Two-photon imaging.** For two photon imaging, after six weeks of virus expression, awake animals in a body tube were head-fixed under a 25X objective (Olympus FV30-AC25W). Imaging was performed with Olympus FVMPE-RS microscopy. Resonant scanner was chosen for calcium dynamic imaging. Functional images (512 × 512 pixels, 0.994 μm/pixel) were collected at 33 Hz or 7 Hz for analyzing fluorescent kinetics. Light source was running at 1,045 nm and the emission light was filtered by a band pass filter (BA575–645). Laser power was set to 105 mW for imaging L2/3 and 135 mW for imaging L5 neurons.

**Passive grating visual stimulation.** Drifting gratings (8 orientations, from 0 to 157.5 degree) visual stimuli movie was generated using the Psychophysics Toolbox3 in MATLAB (Mathworks). It was presented using an 11.6-inch monitor (Raspberry PI), placed 12 cm in front of the center of the left eye of the mouse. Each stimulus trial consisted of a 5-s blank period (uniform grey display at mean luminance) followed by a 2-s drifting sinusoidal grating (0.05 cycles per degree, temporal frequency of 1 Hz, eight randomized different directions). And each stimulus session contains about 90 trials. Visual stimuli were controlled via a pre-programmed pulse table generated by National Instruments DAQ cards in the experiment control PC. And the stimuli signals were synchronized to individual image frames using frame-start pulses provided by FV30S-SW software.

**Analysis of two-photon imaging of the mouse cortex.** Neuronal calcium signals were extracted by Suite2p (Python). ROI detection and signal extraction parameters were uniform for all experiments. And further analysis was performed using MATLAB. Neuropil signals were subtracted from soma signals, with an equation suggested by Suite2p: $F_{corrected} = F_{soma} - 0.7 \times F_{neuropil}$. To screen stimulus response neurons, we first computed trial-average activity of all neurons in each grating directions. Trials were aligned to the stimulus onset. We selected neurons with activity that is significantly different from the baseline in at least one direction (by comparing activity before (−2–0 s) and after (0–2 s) onset, $p < 0.05$, Wilcoxon rank sum test). We than calculated $\Delta F/F_0 = (F - F_0)/F_0$ for response neurons, where $F$ is the instantaneous fluorescence signal and $F_0$ is the average fluorescence in the interval 1 s before the start of the visual stimulus. The baseline fluorescence values in extended Fig 7A were reported using fluorescence intensity at neuronal soma without neuropil correction. For each responsive cell, we defined the preferred stimulus as the stimulus that evoked the maximal $\Delta F/F_0$ amplitude in trial averaged activity (peak values during the 2 s of stimulus presentation). The half-decay time was calculated as the time required for each trace to reach half of its peak value (baseline fluorescence subtracted). It was calculated by interpolation with fitted trial averaged activity using a single exponential function. Only cells for which the trial averaged maximal $\Delta F/F_0$ amplitude was higher than four standard deviations above the baseline signal were included in the analysis. The fraction of cells detected as responsive was calculated as the number of significantly responsive cells over all the cells extracted by Suite2p. OSI of responsive neurons was calculated by comparing the response in the preferred orientation ($R_{pref}$) and that in the orthogonal orientation ($R_{orth}$).

$$OSI = \frac{Rpref - Rorth}{Rpref + Rorth}$$

To obtain pairwise Pearson correlation coefficients, neuronal calcium signals were extracted using Suite2p. The z-score was when computed for each neuronal ROI, both with and without neuropil correction. Pearson correlation coefficients were then calculated from the calcium traces of all possible pairs of neurons across different distances.

For the analysis of spontaneous $Ca^{2+}$ dynamics in the primary visual cortex, images of mice expressing calcium indicators at 6 weeks and 4 months were processed using the CaImAn [56] software suite. Specifically, NormCorre [76] in the CaImAn library was used for piece-wise rigid motion correction of the images and the Constrained Non-negative Matrix Factorization (CNMF) [77] algorithm in CaImAn was used for neuronal signal extraction and spike deconvolution. The peak $\Delta F/F_0$ was determined by identifying the maximum value of the $\Delta F/F_0$ signal derived from the CNMF output, where $F_0$ was defined through a moving windows quantile approach. The peak SNR was computed as the ratio of peak $\Delta F/F_0$ value to the standard deviation of the $\Delta F/F_0$ signal. For S9I Fig, we use isolated single spikes, defined as those without other spikes occurring within 40 frames before or after the event.

**Dual-color two-photon imaging of brainstem**

**Stereotactic NTS injections and surgery.** Stereotactic injections and cranial window implantations were performed as previously described [78]. Briefly, young mice (3–4 weeks old) were anesthetized with ketamine (65 mg/kg) and xylazine (13 mg/kg) and placed in a small animal stereotaxic frame (RWD). Three injections were made at coordinates relative to bregma: 0.36 mm lateral to the midline to the left, and 4.40 mm posterior, 0 mm lateral and 4.65 mm posterior, and 0.12 mm lateral to the left and 4.90 mm posterior. A glass pipette was slowly lowered to a depth of 4.8–5.0 mm for the most rostral injection and 4.3–4.8 mm for the other sites. The viral vectors AAV2/9-Syn-RiboL1-jGCaMP8s-WPRE and AAV2/9-Syn-NES-SomaFRCaMPi-WPRE (titer >1 × $10^{13}$ vg/ml) were mixed in a 1:1 volume-to-volume (v/v) ratio. The mixed viral solution was then injected (150 nL per injection at 2 nl/s) using a Nanoject III injector (Drummond). Then, mice underwent cranial window implantation, anesthetized with isoflurane/urethane and warmed on a custom-made heated platform. The skull over cerebellar lobules VI–IX was carefully removed with a dental drill. The posterior part of the cerebellum, approximately lobules VII–X, was gently aspirated to expose the dorsal surface of the brainstem. A custom-made titanium headpost was mounted on the skull, parallel to the surface of the area postrema, and secured with adhesive cement (C&B Metabond; Parkell) and Kwik-Sil adhesive (World Precision Instruments). Mice recovered for approximately 15 days before functional imaging.

**Visceral stimulation.** Mechanical organ distensions were performed as previously described [78], by inflating a latex balloon (Braintree Scientific; Cat. No. 73–3,479 for the stomach and 73–3,478 for the intestine). For gastric distension, the contents of the stomach were removed, and a small incision was made in the greater curvature of the glandular stomach. The balloon was advanced into the antrum of the stomach and secured with a suture around the incision site. Intestinal distension was achieved by placing the balloon into the duodenal bulb through an incision at the pyloric sphincter and securing it at the sphincter with a suture. For laryngeal stimulation, a 100 µl water bolus was delivered through a syringe connected to the laryngeal cannula with silicon tubing.

**Two-photon NTS imaging.** On the day of imaging, mice were carefully positioned in headpost clamps to ensure parallel alignment of the cranial window with the front lens of the microscope objective. Two-photon calcium imaging was performed using a Nikon resonant-scanning two-photon microscope equipped with a Nikon 16X water-immersion objective (numerical aperture [NA] of 0.6, working distance [WD] of 3 mm) with a zoom factor of 1.5X. The main channel of an Insight X3 laser (Spectra-Physics) was tuned to 920 nm for imaging RiboL1-jGCaMP8s, and the second channel at 1,045 nm was used to image SomaFRCaMPi. Fluorescence emission was filtered with a 500–550 nm bandpass filter for RiboL1-GCaMP8s signals and a 563–588 nm bandpass filter for SomaFRCaMPi signals. Volumetric imaging (1.1 Hz, resolution of 512 × 512 pixels, 525 × 525 µm) was performed, typically consisting of five focal planes 50–80 µm apart.

**Brainstem data analysis for SomaFRCaMPi.** For brainstem data, both SomaFRCaMPi and RiboL1-jGCaMP8s were imaged within the same FOV using separate channels. Multiple z-slices were acquired for each specimen.

Non-rigid motion correction was applied to each $z$-slice using the Normcorre library, followed by the generation of average intensity projections for each corrected slice. ROIs were automatically delineated using Cellpose for each channel, as SomaFRCaMPi and RiboL1-jGCaMP8s expression marked distinct neuron populations. To identify neurons co-expressing SomaFRCaMPi and RiboL1-jGCaMP8s, the intersection-over-union (IOU) was calculated for ROI pairs across channels. Neuron pairs with an IOU threshold of 0.5 or greater and the highest IOU among all potential pairs were identified as the same neuron. Results from individual slices were overlaid to produce comprehensive $z$-axis visualizations. Signal traces for each neuron were calculated as the average of values within the ROIs defined by Cellpose. Fluorescence signals (F) were baseline-corrected using spline fitting to account for photobleaching and served as baseline fluorescence. Normalized traces ($\Delta F/F = (F - \text{baseline})/\text{baseline}$) were used to determine neuronal responsiveness. Neurons were classified as responsive to a given stimulus if their $\Delta F/F$ trace showed a $z$-score exceeding 2.5 within 3 s of stimulus onset.

## One-photon wide-field Ca²⁺ imaging in awake mice

**Virus injection and cranial window implantation.** For cortical injection in one-photon imaging, either AAV2/9-Syn-NES-FRCaMPi-RPL10 or AAV2/9-Syn-NES-FRCaMPi (titer >$10^{13}$ vg/mL, Shanghai Sunbio Medical Biotechnology) were used after being diluted with saline and then mixed with AAV2/9-Syn-GRAB5HT3.0 (final titer $3.00 \times 10^{12}$ vg/mL, WZ Biosciences Inc.) in a 1:1 volume ratio. All of the AAV solutions were colored with 0.1% Fast Green FCF (15,939−54, Nacalai Tesque Inc.). P0–P1 neonatal mice were collected from their cages and slowly dispensed with 1 µl of mixed virus per hemisphere.

Cortex-wide windows were prepared in a similar way according to Ghanbari and colleagues [47]. Briefly, adult mouse (>P50) was anesthetized using 4% isoflurane (induction) and maintained using 1.5–2% on a 37 °C heating pad for the entire surgery procedure. After head-fixing the mouse on a stereotaxic apparatus (RWD Life Science, USA), hair on the scalp was removed using depilatory cream and eyes were covered with erythromycin ophthalmic eye ointment. Then, skin and fascia were removed longitudinally from the occipital bone to nasal bone and laterally between the temporal bones. Before cranial drilling, the skull was cleaned with saline. The bregma and lambda points were identified and the dorsal part of the skull was carefully drilled using a carbide bur (P86-0196, Pearson Dental Supply) attached on a hand-piece (DTL Dental Equipment Manufacture Co., Ltd). The drilling path was made in a closed loop fashion, starting from bregma and lambda, through sagittal suture, extending along both lambdoid and coronal suture to around two-third of the suture length and ending by drilling and joining the terminals of sutures at each side. The skull was then carefully removed using fine surgical forceps. A polymer-based window composed of 3D-printed PMMA frame and PET film was attached to the skull and glued tightly by applying cyanoacrylate (VetBond, 3 M) onto the edge of the frame. A self-tapping screw (F000CE094, Morris Precision Screws and Parts, USA) were implanted to one side of the occipital bone to assist in anchoring the See-Shell to the skull. Dental cement (C&B Metabond, Parkell) was then applied to cover all skull surface close to the window and fix the window firmly to the skull. A custom titanium headplate was then placed above the window and screwed tightly to it. Finally, a layer of denture base resin was applied on top of the dental cement to consolidate the entire window and left for 5–10 min to dry. The mouse was then returned to home cage and allowed to recover for at least 4 days after surgery.

**Wide-field one-photon imaging.** Prior to image acquisition, mice were anesthetized using 2% isoflurane and head-fixed onto a custom disk treadmill. Wide field imaging was performed using an Olympus MVX10 macroscope equipped with a 2×/0.5 numerical aperture (NA) objective lens (MVPLAPO2XC, Olympus), a zoom body (Olympus, MVX-ZB10), and sCMOS cameras (Zyla 4.2 Plus, Andor). The magnification of the zoom body was adjusted to 6.3×. Epi-illumination was provided by a 4-wavelength high-power LED source (LED4D067, Thorlabs), with 470 and 565 nm LEDs adjusted at 45.2 mW and 8.5 mW, respectively. For each image session, 500 frames (2048 × 2048 pixels) were acquired from either S1 or V1 at 10 Hz with 75 ms exposure time during the resting period of the mouse. The curvature of the polymer cranial

window resulted in portions of the FOV being outside of the focal plane. To mitigate this issue, we sampled images from the largest area that remained within the focal plane. Consequently, the representative images shown in Fig 5C illustrate the focused part of the FOV for demonstration purposes. As GECIs were pan-cortically expressed, we expected that fluorescence signals were collected from neurons distributed across cortical layers, but with a larger weight from L2/3 as signals from deeper layers were severely scattered. With the high NA achieved at the selected zoom magnification (i.e., 0.5), the nominal axial resolution of the imaging setup is about 5 μm.

**Analysis of one-photon in vivo calcium imaging data in awake mice.** Images from wide-field calcium imaging were processed using open-access code and methods in the following steps;

**Denoising:** publicly available Python code of SUPPORT [79] was used for denoising the images before signal extraction. Denoising was only performed onto images acquired from mice over 4 months to reduce the noise level dominated by tissue scattering and to aid somatic signal detection. SUPPORT was operated on Pytorch 1.12.1 and CUDA 11.3 with an NVIDIA RTX 3,090 GPU and an Intel Sliver 4212R CPU;

**Identification of neuronal ROIs and extraction of calcium signals:** publicly available MATLAB code of CNMF-E [38] was used for the identification of neurons in 1-photon calcium imaging data. Down-sampling by 4 in $x$ and $y$ dimension and patch-based processing were used to lower the memory requirements for the analysis. AR(1) model was used for the deconvolution of the calcium traces. Ring model was used for identifying the background. ROI detected within or at proximity of blood vessels were excluded. Images with fluorescence change due to significant motion artifacts were excluded;

**Identification of spike events:** spikes were detected by the CNMF-E algorithm with AR1 spike deconvolution model. To identify the calcium signal trace originated from a single spike, spike events where past and future spikes were not within 1 s were chosen. From the chosen spikes, average and standard deviations of those spike signals were calculated;

**Calculation of pairwise Pearson correlation coefficients:** for each ROI detected by CNMF-E algorithm, raw fluorescence traces were extracted from each neuron by calculating the average value under the ROI mask for each frame. Then, Pearson correlation coefficients were then calculated from the calcium traces of all possible pairs of neurons across different distances;

**Calculation of Peak $\Delta F/F_0$ and peak SNR:** To calculate $F0$, B-spline fitting was applied to three frames selected from the fluorescence signal (F) prior to the onset of each identified spike. Peak $\Delta F/F_0$ was then defined as the maximal fluorescence change over baseline identified for the calcium transients of each neuron. Peak SNR was defined as the maximal fluorescence change divided by the standard deviation of $F_0$ as follows.

$$\text{Peak SNR} = \frac{\max(\Delta F)}{std(F0)}$$

**Calculation of fluorescence decay time:** Same as spike identification applied in the "**Identification of spike events**" above, single spike events where past and future spikes were not within 1 s were chosen to extract the $\Delta F/F_0$ trace of each ROI and determine fluorescence decay time. $F_0$ was determined by B-spline fitting to the entire fluorescence trace of each ROI. Full decay time was then calculated using the time between the peak and the first time point that reached the negative $\Delta F/F_0$ of the spike.

## Supporting information

**S1 Fig. Characterized in vitro properties of FRCaMPi indicator. (A)** Absorption spectra for FRCaMPi in the $Ca^{2+}$-bound (sat) and $Ca^{2+}$-free (apo) states at pH 7.20. **(B)** Excitation and emission spectra for FRCaMPi in the $Ca^{2+}$-bound (sat) and $Ca^{2+}$-free (apo) states at pH 7.20. **(C)** Red fluorescence intensity for FRCaMPi in the $Ca^{2+}$-bound (sat) and $Ca^{2+}$-free (sat) states and the $F/F$ dynamic range as a function of pH. **(D)** $Ca^{2+}$ titration curves for FRCaMPi in the absence and presence of 1 mM $MgCl_2$ at pH 7.20. The experimental data were fitted by the Hill equation. **(E)** Photobleaching of

FRCaMPi and control FRCaMP, YTnC, and mEGFP proteins under continuous wide-field imaging using a mercury lamp in the presence of $Ca^{2+}$. **(F)** Fast protein liquid chromatography of FRCaMPi in the presence of $Ca^{2+}$. FRCaMPi (4.5 mg/ml) was eluted in 40 mM Tris-HCl (pH 7.5) and 200 mM NaCl buffer supplemented with 5 mM $CaCl_2$. The molecular weight of FRCaMPi (having a theoretical molecular weight of 48 kDa) was calculated from a linear regression of the dependence of the logarithm of the control molecular weights versus the elution volume. **(C–F)** Three–six replicates were averaged for analysis. Error bars represent the standard deviation.
(TIF)

**S2 Fig. Comparsion of FRCaMP and FRCaMPi fluorescence response in neurons. (A)** Representative images from neuronal culture expressing FRCaMPi. **(B)** Peak $\Delta F/F_0$ of FRCaMP and FRCaMPi at 87 Hz, 300 pulses. **(C)** Half decay time of FRCaMP and FRCaMPi at 87 Hz, 300 pulses (25 cells each, from 2 independent cultures). Mann–Whitney U test: not significant. Box indicates the median and 25–75th percentile range, and the whiskers represent 1.5 times of interquartile range. The quantitative data presented in this figure can be found in S2 Data.
(TIF)

**S3 Fig. Characterization of GECIs in cultured neurons via electric field stimulation. (A)** Representative trace of concurrent voltage and calcium recording in neuron co-expressing GCaMP6f and SomArchon. Pulse numbers 1, 2, 3, 5, 10, 20, and 40 were evoked with 1 ms width and frequency of 83 Hz. 40–50 V voltage was applied. An extra spike was detected before 3 AP stimulation. Images were sampled at 1 kHz for NIR and 25 Hz for FITC channel. **(B)** Representative standard deviation projection image of neuron from **(A)** showing the maximal fluorescence across all the frames. **(C)** Cropped voltage traces showing the exact spike number from voltage recording in **(A)**. The numbers of external electrical pulses applied were aligned with the number of spikes observed, indicated by fluorescence of SomArchon.
(TIF)

**S4 Fig. Characterization of baseline brightness and photobleaching of red GECIs in primary hippocampal neurons. (A)** Representative images of neuron co-expressing red GECI (TRITC) and emiRFP670 (NIR). ($n$ = 5 from two independent neuronal cultures). **(B)** Representative images of neuron co-expressing soma-localized version of red GECI (TRITC) and emiRFP670 (NIR) (top). Scale bar, 50 μm. ($n$ = 5 from two independent neuronal cultures). **(C)** Baseline brightness of red GECIs characterized in transduced primary hippocampal neurons by side-by-side comparison. K-GECO1: $n$ = 55 cells from 3 wells; XCaMP-R: $n$ = 31 cells from 2 wells; jRGECO1a: $n$ = 59 cells from 3 wells; FRCaMPi: $n$ = 87 cells from 4 wells, two independent cultures. Kruskal–Wallis multiple-comparison test. Data is normalized to the median of the jRGECO1a values. **(D)** Baseline brightness of red GECIs compared to that of their soma-localized version. Data is normalized to the median of the soma-localized GECI values. Statistics in d for are the same as in c and Fig 2H. **(E)** 5 min recording of fluorescence trace from neurons expressing red GECI before applying electrical field stimuli. $n$ = 34 cells for K-GECO1, $n$ = 31 cells for XCaMPR, $n$ = 49 cells for jRGECO1a, $n$ = 14 cells for FRCaMPi. **(F)** As in f[To AU: Please check whether the part label "f" referred here is correct. Please confirm.] but for soma-localized red GECIs. $n$ = 35 cells for SomaK-GECO1, $n$ = 51 cells for SomaXCaMP-R, $n$ = 59 cells for SomajRGECO1a, $n$ = 28 cells for SomaFRCaMPi. **(G)** Peak $\Delta F/F_0$ of red GECI compared to that of their corresponding soma-localized version as a function of pulses K-GECO1: $n$ = 54 cells from 3 coverslips, 2 independent cultures; SomaK-GECO1: $n$ = 61 cells from 4 coverslips, 3 independent cultures; XCaMP-R: $n$ = 24 cells from 2 coverslips, 2 independent cultures; SomaXCaMP-R: $n$ = 64 cells from 4 coverslips, 3 independent cultures; jRGECO1a: $n$ = 65 cells from 4 coverslips, 3 independent cultures; SomajRGECO1a: $n$ = 100 cells from 6 coverslips, 4 independent cultures. The quantitative data presented in this figure can be found in S2 Data.
(TIF)

**S5 Fig. Fiber photometry recording of water- and air puff-induced neuronal activity in CeA. (A)** FRCaMPi was expressed and optical fibers were implanted bilaterally in CeA (CeAL: Left CeA; CeAR: Right CeA. **(B)** Task schematics showing the time of stimulus onset. **(C)** Licking events of a mouse in 20 trials. **(D)** FRCaMPi signal recorded in CeA in response to water. **(E)** FRCaMPi signal recorded in CeA in response to water air puff. The quantitative data presented in this figure can be found in S2 Data.
(EPS)

**S6 Fig. Screening of soma-localized peptide fusions FRCaMPi in primary hippocampal neuronal culture.** We made fusions between FRCaMPi and the peptides known to facilitate somatic localization. We then screened those fusions with a focus on their functionality: cytotoxicity (assessed as total number of functional cells obtained from trans-fected neuronal culture), **(A)** single-trial single-cell optical traces from representative neurons expressing **(B)** fluorescence changes (peak/max $\Delta F/F_0$) and **(C)** SNR in response to evoked extracellular electrical stimuli ($n$ = X, Y, Z neurons from 1 matching culture for, respectively). We found that ribo-tagged FRCaMPi (FRCaMPi-RPL10a) demonstrates the least toxicity, highest peak $\Delta F/F_0$ on average with SNR comparable to FRCaMPi. We thereby pursued the ribo-tagged construct for more detailed characterization, naming it SomaFRCaMPi. The quantitative data presented in this figure can be found in S2 Data.
(TIF)

**S7 Fig. In vivo performance of SomajRGECO1a in neurons in zebrafish larvae. (A)** In vivo confocal images showing fore-, mid- and hindbrain region of live F1 zebrafish larva expressing SomajRGECO1a and SomaFRCaMPi in neurons driven by the HuC promoter. The dynamic range was adjusted identical for each image to facilitate visual comparison of brightness. SomajRGECO1a was imaged at 2 mW whereas SomaFRCaMPi was imaged using 0.8 mW laser power. The white dashed box indicates local area of SomajRGECO1a image adapted enhanced contrast to aid visualization. The baseline brightness of SomajRGECO1a in F1 is approximately three times dimmer compared to that of SomaFR-CaMPi. **(B)** Representative confocal images showing fluorescence response of optical tectum in larva expressing Soma-jRGECO1a before and during looming stimulation periods. The laser power used to image SomajRGECO1a in F1 fish (0.74 mW/mm$^2$) was 10 times to that of jRGECO1a, FRCaMPi and SomaFRCaMPi in F3 (0.076 mW/mm$^2$). **(C)** Representative fluorescence traces of soma and neuropil region in optical tectum of larva expressing SomajRGECO1a during looming stimulation. **(D)** Peak $\Delta F/F_0$ and peak SNR of the fluorescence response at neuronal soma of Somaj RGECO1a-expressing fish during looming stimulation. $n$ = 24 cells from 4 fish. Scale bar, 50 μm.
(TIF)

**S8 Fig. In vivo performance of FRCaMPi and SomaFRCaMPi in neurons in zebrafish larvae. (A)** Baseline bright-ness of larval fish expressing jRGECO1a or FRCaMPi variants $n$ = 13 fish for jRGECO1a, $n$ = 14 fish for FRCaMPi, $n$ = 22 fish for SomaFRCaMPi. c. Half rise (left) and decay (right) time of red GECIs in response to looming stimuli. **(B)** Representative single plane confocal images of periventricular cell bodies in optic tectum of fish expressing jRGE-CO1a, FRCaMPi or SomaFRCaMPi. Decreased neuropil fluorescence and clearer neuronal cell bodies was found in SomaFRCaMPi-expressing larval zebrafish. **(C)** Larval zebrafish were mounted into agarose gel and calcium response was evoked by electrical stimuli delivered by platinum sheet electrodes and a voltage amplifier. Neurons activity in the superior medulla oblongata were recorded. **(D)** Averaged $\Delta F/F_0$ Ca$^{2+}$ transients from neurons in the superior medulla oblongata, responding to electrical field stimulation. Averaged traces are shown as mean ± s.e.m, indicated by solid lines and shaded area, respectively. **(E)** Peak $\Delta F/F_0$ (left) and peak SNR (right) of neurons expressing jRGECO1a, FRCaMPi or SomaFRCaMPi in the superior medulla oblongata, responding to electrical field stimulation. One-way ANOVA test was used followed by a post hoc Tukey test. The Shapiro–Wilk test was used for the normal distribution test. **(F)** Half rise (left) and decay (right) time of neurons expressing jRGECO1a or FRCaMPi variants in the superior medulla oblongata,

responding to electrical field stimulation. A single circular ROI containing 4–5 neurons was selected for each fish. A total of 8 fish were recorded for jRGECO1a, FRCaMPi and SomaFRCaMPi each. Statistics in **d–e** are the same as in **f**. **(G)** Half rise (left) and decay (right) time of the fluorescence response during looming stimulation (jRGECO1a: $n = 18$ cells; FRCaMPi: $n = 18$ cells; SomaFRCaMPi: $n = 17$, 3 fish each) Kruskal–Wallis ANOVA was used followed by a post hoc Dunn's multiple comparison test. In dot plots, dots are individual data points; the line denotes the median. Bar chart is shown as mean ± s.e.m. Box indicates the median and 25–75th percentile range, and the whiskers represent 1.5 times of interquartile range. The quantitative data presented in this figure can be found in S2 Data. Created in BioRender, https://BioRender.com/j92q145, https://BioRender.com/q72h354.
(TIF)

**S9 Fig. In vivo characterization in mice V1 cortex. (A)** Box plot of baseline fluorescence for segmented ROIs of V1 L2/3 responsive neurons. Central line inside the box: median. Box: interquartile range. Whiskers: maximum and minimum. Outliers: individual data points. **(B)** Example fluorescence traces of 3 cells responded to the preferred grating stimuli. Solid line, mean. Shaded area represents SEM. Trials are aligned to the stimulus onset that is indicated by the solid arrow. **(C)** Box plot of half-decay time of the fluorescence response after the end of the visual stimulus. **(A)** or **(C)** used the same cells in Fig 5E or 5F, Kruskal–Wallis ANOVA test with Dunn's multiple-comparison was used). **(D)** Cumulative distribution function of the peak SNR for V1 L2/3 neurons in response to the preferred stimuli. **(E)** Same as **(D)** but for V1 L5 neurons. **(D)** or **(E)** used same cells in Fig 5E or 5F, two-sample Kolmogorov–Smirnov test was used). **(F)** Histogram of orientation selectivity index (OSI) of V1 L2/3 responsive neurons. Neurons exhibited OSI higher than 1 (jRGECO1a, 0.90%, FRCaMPi, 2.1%, SomaFRCaMPi, 0.21%) were excluded here. **(G)** Box plot of OSI values of V1 L2/3 responsive neurons. The distributions of OSI for the three indicators are similar (median: jRGECO1a = 0.42, FRCaMPi = 0.45, SomaFRCaMPi = 0.42). Kruskal–Wallis ANOVA test with Dunn's multiple-comparison was used. **(H)**. Representative single-plane images of V1 neurons expressing jRGECO1a for 6 or 16 weeks, and SomaFRCaMPi for 6 or 16 weeks (jRGECO1a week 6: $n = 14$ FOVs from 3 mice; jRGECO1a week 16: $n = 6$ FOVs from one mouse; SomaFRCaMPi week 6: $n = 10$ FOVs from 2 mice; SomaFRCaMPi week 16: $n = 6$ FOVs from one mouse). **(I)** Example deconvolved fluorescence traces of spontaneous spike (jRGECO1a week 6: $n = 314$ spikes; jRGECO1a week 16: $n = 69$ spikes; SomaFRCaMPi week 6: $n = 763$ spikes; SomaFRCaMPi week 16: $n = 366$ spikes). Solid or dotted line, mean. The shaded area represents SEM. **(J)** Box plot of peak SNR values of spontaneous activity from V1 L2/3 neurons (median: jRGECO1a–week 6 = 8.65, $n = 1,338$ cells; jRGECO1a–week 16 = 7.89, $n = 580$ cells; SomaFRCaMPi-week 6 = 8.87, $n = 1,242$ cells; SomaFRCaMPi-week 16 = 8.48, $n = 1,748$ cells). **(K)** Box plot of peak ΔF values of V1 L2/3 neurons spontaneous activity (median: jRGECO1a–week 6 = 0.28, $n = 1,338$ cells; jRGECO1a–week 16 = 0.21, n = 580 cells; SomaFRCaMPi-week 6 = 0.41, $n = 1,242$ cells; SomaFRCaMPi-week 16 = 0.29, $n = 1,748$ cells). Mann–Whitney U test was used in **(J)** and **(K)**. The quantitative data presented in this figure can be found in S2 Data.
(EPS)

**S10 Fig. Two-photon imaging of mouse parabrachial nucleus neurons in acute brain slice. (A)** ROI segmentation of SomaFRCaMPi-expressing neurons in the parabrachial nucleus. Left: Representative image of single FOV (top) expressing FRCaMPi, blue square is magnified in the bottom panel. Right: Corresponding ROI masks extracted using Cellpose. **(B)** Similar to **(A)**, but in SomaFRCaMPi-expressing neurons. **(C)** ROI number extracted per FOV for SomaFRCaMPi and FRCaMPi experiments (SomaFRCaMPI; 2 mice, 5 brain slices, 22 FOVs. FRCaMPi; 2 mice, 4 brain slices, 18 FOVs). **(D)** Example heatmaps for 15 min of activity from two example brain slices (4 simultaneously acquired FOVs per slice). **(E)** Example traces demonstrating calcium waveform from **(D)**. The quantitative data presented in this figure can be found in S2 Data.
(EPS)

**S11 Fig. SomaFRCaMPi and FRCaMPi expression in brains of mice for wide-field one photon imaging. (A)** Top: coronal section demonstrating that FRCaMPi are expressed pan-cortically in S1 (left) and V1 (right) region. Bottom: representative image showing L2/3 neurons in V1 with FRCaMPi (magenta) and Nissl staining (green). **(B)** Same as a but for a mouse brain expressing SomaFRCaMPi. Adeno-associated viruses encoding FRCaMPi or SomaFRCaMPi were delivered by intracerebral ventricular injection into neonatal mice bilaterally. **(C)** Ratio of FRCaMPi- or SomaFRCaMPi-expressing neurons to Nissl positive neurons quantifies the labeling density at region S1 or V1. S1: $n = 6$ slides from 2 FRCaMPi mice, $n = 8$ slides from 3 SomaFRCaMPi mice; V1: $n = 8$ slides from 3 FRCaMPi mice, $n = 6$ slides from 2 SomaFRCaMPi mice. **(D)** Pearson correlation coefficient of calcium dynamics between pairs of neurons within the range of 0–200 μm, 200–400 μm and 400–800 μm. Statistics is the same as in Figs 5K and l. Mann–Whitney U test was used. Scale bar, 25 μm, 100 μm and 500 μm. Bar chart is shown as mean ± s.e.m. Box indicates the median and 25–75th percentile range, and the whiskers represent 1.5 times of interquartile range. Pearson correlation coefficient is plotted as bar to aid demonstration of data distribution. See S6 Table for more details. The quantitative data presented in this figure can be found in S2 Data. (TIF)

**S1 Movie. Live cell wide-field imaging of HeLa cells expressing FRCaMPi during drug-induced calcium responses.** Cells were imaged for 16 min (0.33 μm/pixel and acquired at 4.3 Hz). Scale bar, 50 μm. (MP4)

**S2 Movie. Dual-color mesoscopic wide-field imaging of mouse dorsal cortex co-expressing gGRAB$_{5-HT3.0}$ and SomaFRCaMPi during KA-induced seizure.** Imaging conditions: 3.74 × 3.74 mm$^2$ (576 × 576 pixels, 4× binning), acquired at 20 Hz. Scale bar, 500 μm. (MP4)

**S3 Movie. Single-plane confocal imaging of optical tectum in larval zebrafish expressing FRCaMPi, jRGECO1a, or their soma-localized variant in response to looming stimuli.** Imaging conditions: 512 × 512 pixels, 1.243 μm/pixel, acquired at 3 Hz. Scale bar, 100 μm. (MP4)

**S4 Movie. Single-plane 2-photon imaging for FRCaMPi, jRGECO1a or their soma-localized variant from L2/3 neurons of primary visual cortex in awake mouse.** Imaging conditions: 512 × 512 pixels, 0.994 μm/pixel, acquired at 7 Hz. Scale bar, 100 μm. (MP4)

**S5 Movie. Wide-field one-photon imaging of FRCaMPi and SomaFRCaMPi in the primary visual and somatosensory cortex of the awake mouse.** Imaging conditions: 1.06 × 1.06 mm$^2$ (2,048 × 2,048, 0.519 μm/pixel), acquired at 10 Hz. Scale bar, 250 μm. (MP4)

**S1 Table. List of amino acid sequences for peptide screening.** (DOCX)

**S2 Table. List of primers used in this study.** (DOCX)

**S3 Table. Summarized In vitro properties of FRCaMPi compared to FRCaMP.** (DOCX)

**S4 Table. Statistics on FRCaMPi characterization in primary hippocampal neurons.** (DOCX)

**S5 Table. Statistics on SomaFRCaMPi characterization in primary hippocampal neurons.**
(DOCX)

**S6 Table. Statistics on FRCaMPi and SomaFRCaMPi characterization in wide-field one-photon cortical imaging of awake mice.**
(DOCX)

**S1 Text. Statistics on FRCaMPi and SomaFRCaMPi characterization in wide-field one-photon cortical imaging of awake mice.**
(DOCX)

**S1 Data. Source data.**
(XLSX)

**S2 Data. Source data.**
(XLSX)

## Acknowledgements

We thank Y.Z., G.Z. and M.W. from Qionghai Dai lab, Tsinghua University for their help and advice with FRCaMPi and SomaFRCaMPi in vivo imaging under wide-field. We thank W.G. from Youjun Wang lab, Beijing Normal University for suggestions on HeLa cell culture experiment. We also thank the Laboratory Animal Resources Center and the Microscopy Core Facility in Westlake University. Henrietta Lacks, and the HeLa cell line that was established from her tumor cells without her knowledge or consent in 1951, have made significant contributions to scientific progress and advances in human health. We are grateful to Henrietta Lacks, now deceased, and to her surviving family members for their contributions to biomedical research. C.R. acknowledges S. Wang and T. Huang for assisting two-photon calcium imaging surgery.

## Author contributions

**Conceptualization:** Shihao Zhou, Qiyu Zhu, Oksana M. Subach, Yu Mu, Fedor V. Subach, Kiryl D Piatkevich.

**Data curation:** Shihao Zhou, Qiyu Zhu, Minho Eom, Shilin Fang, Oksana M. Subach, Chen Ran, Jonnathan Singh Alvarado, Praneel S. Sunkavalli, Yuanping Dong, Fedor V. Subach, Kiryl D Piatkevich.

**Formal analysis:** Shihao Zhou, Qiyu Zhu, Minho Eom, Shilin Fang, Oksana M. Subach, Chen Ran, Jonnathan Singh Alvarado, Praneel S. Sunkavalli, Yuanping Dong, Jiewen Hu, Zhiyuan Wang, Fedor V. Subach, Kiryl D Piatkevich.

**Funding acquisition:** Kiryl D Piatkevich.

**Investigation:** Shihao Zhou, Qiyu Zhu, Minho Eom, Oksana M. Subach, Yangdong Wang, Fedor V. Subach.

**Methodology:** Shihao Zhou, Qiyu Zhu, Minho Eom, Shilin Fang, Oksana M. Subach, Chen Ran, Jonnathan Singh Alvarado, Praneel S. Sunkavalli, Yuanping Dong, Yangdong Wang, Hanbin Zhang, Xiaoting Sun, Fedor V. Subach, Kiryl D Piatkevich.

**Project administration:** Shihao Zhou, Kiryl D Piatkevich.

**Resources:** Chen Ran, Tao Yang, Yu Mu, Young-Gyu Yoon, Zengcai V. Guo, Fedor V. Subach, Kiryl D Piatkevich.

**Software:** Shihao Zhou, Qiyu Zhu, Minho Eom, Shilin Fang.

**Supervision:** Tao Yang, Yu Mu, Young-Gyu Yoon, Zengcai V. Guo, Fedor V. Subach, Kiryl D Piatkevich.

**Validation:** Shihao Zhou, Qiyu Zhu, Minho Eom, Shilin Fang, Oksana M. Subach, Chen Ran, Jiewen Hu.

**Visualization:** Shihao Zhou, Qiyu Zhu, Minho Eom, Shilin Fang, Jonnathan Singh Alvarado, Praneel S. Sunkavalli, Yuanping Dong, Fedor V. Subach, Kiryl D Piatkevich.

**Writing – original draft:** Shihao Zhou, Kiryl D Piatkevich.

**Writing – review & editing:** Shihao Zhou, Qiyu Zhu, Minho Eom, Shilin Fang, Oksana M. Subach, Yu Mu, Young-Gyu Yoon, Zengcai V. Guo, Fedor V. Subach, Kiryl D Piatkevich.

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
