## [Editor Report · Decision Letter 0]

24 Jan 2025

Dear Dr Piatkevich, 

Thank you for submitting your manuscript entitled "A Sensitive Soma-localized Red Fluorescent Calcium Indicator for Multi-Modality Imaging of Neuronal Populations In Vivo" for consideration as a Research Article by PLOS Biology. I am sorry for the slight delay in getting back to you as we consulted with an academic editor about your submission.

Your manuscript has now been evaluated by the PLOS Biology editorial staff, as well as by an academic editor with relevant expertise, and I am writing to let you know that we would like to send your submission out for external peer review. 

IMPORTANT: After discussions with the rest of the team, we think your manuscript would be a better fit as a 'Methods and Resources' article at the journal. During resubmission (details below), I would be grateful if you could please tick 'Methods and Resources' as the article type in the dropdown menu. 

Once your full submission is complete, your paper will undergo a series of checks in preparation for peer review. After your manuscript has passed the checks it will be sent out for review. To provide the metadata for your submission, please Login to Editorial Manager (https://www.editorialmanager.com/pbiology) within two working days, i.e. by Jan 26 2025 11:59PM.

Kind regards,

Richard

Richard Hodge, PhD

rhodge@plos.org

PLOS

---

## [Decision Letter · Decision Letter 1]

20 Feb 2025

Dear Dr Piatkevich,

Thank you for your patience while we considered your revised manuscript "A Sensitive Soma-localized Red Fluorescent Calcium Indicator for Multi-Modality Imaging of Neuronal Populations In Vivo" for publication as a Methods and Resources at PLOS Biology. This revised version of your manuscript has been evaluated by the PLOS Biology editors, the Academic Editor and the original reviewers.

Based on the reviews and on our Academic Editor's assessment of your revision, we are likely to accept this manuscript for publication, provided you satisfactorily address the following data and other policy-related requests:

* We would like to suggest a different title to improve its accessibility for our broad audience: 

A sensitive soma-localized red fluorescent calcium indicator for in vivo imaging of neuronal populations at single-cell resolution

* Please add the links to the funding agencies in the Financial Disclosure statement in the manuscript details.

* DATA POLICY:

Regardless of the method selected, please ensure that you provide the individual numerical values that underlie the summary data displayed in the following figure panels as they are essential for readers to assess your analysis and to reproduce it: 1CHI, 2HIJKOP, 3BDHIJK, 4DFJL, 5EFGHIJM, S2BC, S3AD, S4CD, S5AEFG, S6ACGJK and S8CD. 

* CODE POLICY

We expect to receive your revised manuscript within two weeks. 

*Published Peer Review History*

*Press*

Sincerely,

Christian

Christian Schnell

Senior Editor

PLOS Biology

cschnell@plos.org 

on behalf of 

Richard Hodge, PhD

Senior Editor

rhodge@plos.org

PLOS Biology

Reviewer remarks:

Reviewer #1: The authors have addressed all my concerns. The addition of the GCaMP experiments have significantly improved the message of the paper and it is suitable for publication in Plos biology in its current form.

Reviewer #2: In response to the previous round of reviews, the authors have added experiments to compare somaFRCaMPi with soma-localized GCaMP8s using 2P imaging in mouse brain, and shown that they perform similarly. In addition to other edits they have also added discussion about potential photoswitching of their sensor that may limit its utility. Although the authors have still not experimentally addressed the important potential limitations of this type of sensor (photoswitching, incomplete maturation in vivo) or experimentally demonstrated any application highlighting why someone would use a red GECI (multiplexed functional imaging, deep tissue imaging, combination with channelrhodopsin), which significantly limits the impact of this work, I will still recommend publication because they demonstrate some situations where their sensor does work and it appears to be the current best-in-class red FP-based GECI.

---

## [Editor Report · Decision Letter 2]

7 Mar 2025

Dear Kiryl,

On behalf of my colleagues and the Academic Editor, Polina Lishko, I am pleased to say that we can accept your manuscript for publication, provided you address any remaining formatting and reporting issues. These will be detailed in an email you should receive within 2-3 business days from our colleagues in the journal operations team; no action is required from you until then. Please note that we will not be able to formally accept your manuscript and schedule it for publication until you have completed any requested changes.

PRESS

Best wishes, 

Richard

Richard Hodge, PhD

rhodge@plos.org

PLOS
